# DT-Pro: Proactive Decision Transformers with Implicit Latent Space Planning

## Abstract

Decision Transformers (DTs) address decision making problems through sequence modeling and have achieved surprisingly strong results. However, DTs still struggle in long-horizon tasks due to their poor planning ability. Existing works have demonstrated that subgoal prediction helps to guide DTs' decision making in complex and long-horizon tasks. However, explicit planning via subgoal prediction suffers from suboptimality, inefficiency and instability. In this paper, we present DT-Pro, a variant of DT that enhances its planning ability by integrating a natural implicit planning step into sequence modeling. Compared with explicit planning via subgoal prediction, the implicit planning works by inferring a latent plan from a structured plan space. Through this way, DT-Pro enables high-quality adaptive plan generation and efficient stepwise replanning with only a marginal increase in the computational cost. Extensive experimental results show that DT-Pro achieves strong performance on a variety of widely used control and navigation benchmarks.

## 1 Introduction

Offline Reinforcement Learning learns policies from a fixed dataset without interaction with the environment. This setting faces fundamental challenges such as value overestimation and uncontrollable error propagation. In recent years, sequence modeling methods such as Decision Transformer (Chen et al., 2021) and Trajectory Transformer (Janner et al., 2021) have emerged as alternative solutions to offline RL. For example, DT constructs sequences of returns-to-go (RTGs), states and actions, and leverages a Transformer to predict actions in an autoregressive manner. DT has proven effective in a variety of decision making tasks, showing its potential in many practical applications.

Despite its advantages, DT faces several major limitations in long-horizon tasks. First, DT predicts the next action solely based on historical sequence, which often leads to myopic strategies. In other words, the auto-regressive architecture essentially limits its planning ability. Second, DT heavily relies on expert trajectories to learn good policies while expert trajectories are often sparse in long-horizon tasks, given that the success rate often drastically decreases as the horizon length grows. Third, with relatively short model lengths, DT struggles to capture the whole context of long sequences. Therefore, improving DT's planning ability is an urgent but non-trivial task.

Recently, several variants of DT aim to improve its foresight by incorporating future information into sequence modeling. For example, ADT predicts k-step further subgoals and conditions action prediction on the subgoals Ma et al. (2023). Similarly, Waypoint-Transformer introduces waypoints as subgoals to improve the utilization of suboptimal trajectories Badrinath et al. (2023b). However, both subgoals and waypoints are manually selected from future information, which reflects only partial and suboptimal information for future plans. PDT leverages the coding of future trajectories to assist the current action prediction Xie et al. (2023). Nonetheless, PDT simply encodes the whole future trajectories, which contain a large amount of noisy and irrelevant information that may mislead the current decision making. Overall, these works demonstrate that future information indeed helps DT with achieving the long-term goal, but they lack sufficient modeling and exploiting of future information, thus may result in suboptimal or even harmful future plans.

In this paper, we propose DT-Pro, a novel sequence modeling framework that aims to improve DT's planning ability while keeping its efficiency. The core idea of DT-Pro is to condition DT's action

prediction on an informative latent representation of the future plans, so that to make it proactive. The overall framework of DT-Pro consists of three modules. First, the *plan search module* searches for a future plan for any given state by looking ahead to critical subgoals in the trajectory. The critical subgoals are selected based on the decaying returns-to-go (RTGs) and can be regarded as important milestones in a plan. Second, the *plan coding module* learns a compact latent space to represent the plans. This module includes a simple plan encoder and a Transformer-based plan decoder, which are trained by autoregressively predicting the next critical subgoals under a contrastive regularizer. Finally, the *action prediction module* generates proactive actions based on the latent plan inferred from the plan encoder. Note that DT-Pro does not require plan decoding during testing, which allows efficient stepwise replanning with only one step of inference.

Unlike DT, DT-Pro adopts a two-stage training procedure. At the first stage, the plan encoder and the decoder are jointly trained using the sequences of the critical subgoals found by the plan search module. At the second stage, a plan conditioned DT policy is trained with the plan encoder frozen.

Our core contributions are as follows.

- We present DT-Pro, a novel variant of DT which significantly enhances its planning ability by introducing an efficient and effective implicit planning mechanism.
- We develop a novel RTG-based plan search algorithm to automatically find critical subgoals, which significantly improves the optimality of future plans. We also show that our plan search algorithm can be easily extended to sparse reward environments.
- We propose a plan coding framework that aims to learn a structured latent plan space, which enables fast inference of informative latent plans at each time step. We show that the latent representation of plans also improves the policy's robustness against noise in states.
- We conduct extensive experiments to demonstrate the superior performance of DT-Pro, compared with both offline RL methods and recent variants of DT in multiple widely used control and navigation benchmarks.

## 2 PRELIMINARIES

### 2.1 OFFLINE REINFORCEMENT LEARNING

Consider a Markov Decision Process (MDP) defined by the tuple $M = (\mathcal{S}, \mathcal{A}, P, r, \gamma)$, where $\mathcal{S}$ and $\mathcal{A}$ are the state and action spaces, $P$ is the transition function, $r$ is the reward function, and $\gamma \in [0, 1]$ is a discount factor. In online RL setting, an agent interacts with the environment by generating trajectories $\tau = (S_0, a_0, r_0, \ldots, S_T, a_T, r_T)$, from which to learn a policy $\pi$ that maximizes the expected return: $\mathbb{E}^\pi \left[ \sum_{t=0}^T \gamma^t r_t \right]$. In the offline RL setting, the agent learns from a fixed dataset $\mathcal{D} = \{\tau_i\}_{i=1}^n$ collected by some unknown behavior policies. Offline RL faces fundamental challenges in estimating the Q-values and preventing the error propagation during training.

### 2.2 DECISION TRANSFORMER

Decision Transformer offers an alternative approach to offline RL. By treating the decision making as a sequence modeling problem, DT focuses on learning the correlations between returns-to-go (RTGs), states and actions. With minimal changes to the Transformer architecture, DT represents trajectories as sequences of states, actions and RTGs:

$$\tau = (\hat{R}_0, S_0, a_0, \hat{R}_1, S_1, a_1, \ldots, \hat{R}_T, S_T, a_T),$$

where $\hat{R}_t = \sum_{i=t}^T r_i$ denotes the return-to-go from timestep $t$. DT learns to autoregressively predict the actions based on states and RTGs, saving the efforts in estimating Q-values. However, the architecture of DT limits its ability to plan for the future, resulting in poor performance and low data efficiency in many long-horizon tasks.

## 3 RELATED WORKS

### 3.1 OFFLINE RL AS SEQUENCE MODELING PROBLEM

Offline RL methods face fundamental challenges in estimating Q-values and preventing error propagation. In recent years, sequence modeling methods, such as DT, have emerged as alternative

solutions to offline RL (Chen et al., 2021; Furuta et al., 2021). Following its success, many variants of DT have been proposed to further improve its performance. For example, Q-DT (Yamagata et al., 2023) attempts to guide strategies toward high-value behaviors; EDT (Wu et al., 2023) dynamically adjusts the context length to enable the model to be both stable and adaptive. ODT (Zheng et al., 2022) extends DT to an online setting by introducing a novel replay buffer and an exploration mechanism. However, although the unified sequential framework reduces the complexity of modeling, it sacrifices the planning ability and limits its performance in long-horizon tasks.

## 3.2 DECISION MAKING WITH PLANNING

Planning plays an important role in decision making, which has been demonstrated in both RL and Large Language Model (LLM) settings. In the RL setting, traditional Q-learning or policy gradient methods do not explicitly learn how to plan. To this end, many hierarchical RL methods are proposed, where high-level policies output subgoals and low-level policies output actions based on the subgoals (Nachum et al., 2018; Levy et al., 2017; Pateria et al., 2021; Jothimurugan et al., 2021; Kim et al., 2021). In this way, the performance of RL-based agents in complex environments is substantially improved. Planning is also important for LLM-based agents Wu et al. (2024). Popular methods such as Chain-of-Thought (Wei et al., 2022), Tree-of-Thought (Yao et al., 2023a), ReAct (Yao et al., 2023b) and AgentGen (Hu et al., 2025) have demonstrated that thinking and planning are essential to improve LLMs' ability to solve complex problems. In addition, TAP (Jiang et al., 2022) uses discrete latent variables to accelerate long trajectory generation. However, as a class of lightweight and practical decision models, DT and its variants still suffer from poor planning ability.

## 3.3 PREDICTIVE CODING FOR DECISION TRANSFORMERS

The original architecture of DT focuses on predicting the next action based only on historical sequence, which often leads to overly myopic strategies. Some recent works try to leverage future information to guide current decision steps. For example, ADT (Ma et al., 2023) borrows the idea from hierarchical RL and introduces a high-level policy for predicting future subgoals, which are then used to prompt the low-level DT. Going further, PDT (Xie et al., 2023) uses future segments of historical trajectories as pre-training targets to enhance the model's understanding of long-term behavioral patterns. Similarly, PCDT (Luu et al., 2024) designs an encoding mechanism based on future state prediction to improve the model's action generation performance in multi-stage tasks. In addition, Waypoint-Transformer (Badrinath et al., 2023b) introduces a waypoint network to predict future goals or rewards that may assist the current decision making. Although the above methods have demonstrated the value of using future information, they still lack a systematic modeling and exploitation of the future information. By contrast, DT-Pro learns a structured plan space using the identified explainable future subgoals, facilitating the optimal plan inference during execution.

## 4 METHODOLOGY

In this section, we present DT-Pro, a novel decision sequence modeling framework that enhances the planning ability and efficiency of DT in long-horizon tasks. DT-Pro introduces an implicit planning mechanism, where a latent plan is inferred from a pretrained plan encoder before action generation. Specifically, DT-Pro consists of three key components: the plan search module, the plan coding module, and the action prediction module. In addition, DT-Pro adopts a two-stage training scheme to stabilize the learning process. The overall architecture of DT-Pro is illustrated in Figure 1.

## 4.1 PLAN SEARCH: SEARCHING CRITICAL SUBGOALS BASED ON RTG DECAY

To support long-horizon planning, we introduce the Plan Search module that extracts a sequence of critical subgoals in the trajectory. Instead of selecting future subgoals based on a pre-determined interval, the Plan Search module identifies critical subgoals based on a decaying view of RTG. The motivation is that, following a good plan will gradually reduce the RTG, and milestones can be marked when the RTG reduces to certain levels.

Given an input trajectory $\tau = (\hat{R}_t, S_t, a_t)_{t=0}^{T}$, our goal is to identify $n$ critical subgoals for each state in the trajectory. We start by constructing a decay factor set $\lambda = \left\{ \lambda_i = \frac{n-i}{n} \mid i = 1, 2, ..., n \right\}$, so that we can compute a target value $\hat{R}_t^{gi} = \lambda_i \cdot \hat{R}_t$ at time $t$. For each $\hat{R}_t^{gi}$, we scan forward in the trajectory and identify the future timestep $t' > t$ such that the RTG $\hat{R}_{t'}$ is closest to $\hat{R}_t^{gi}$. The corresponding

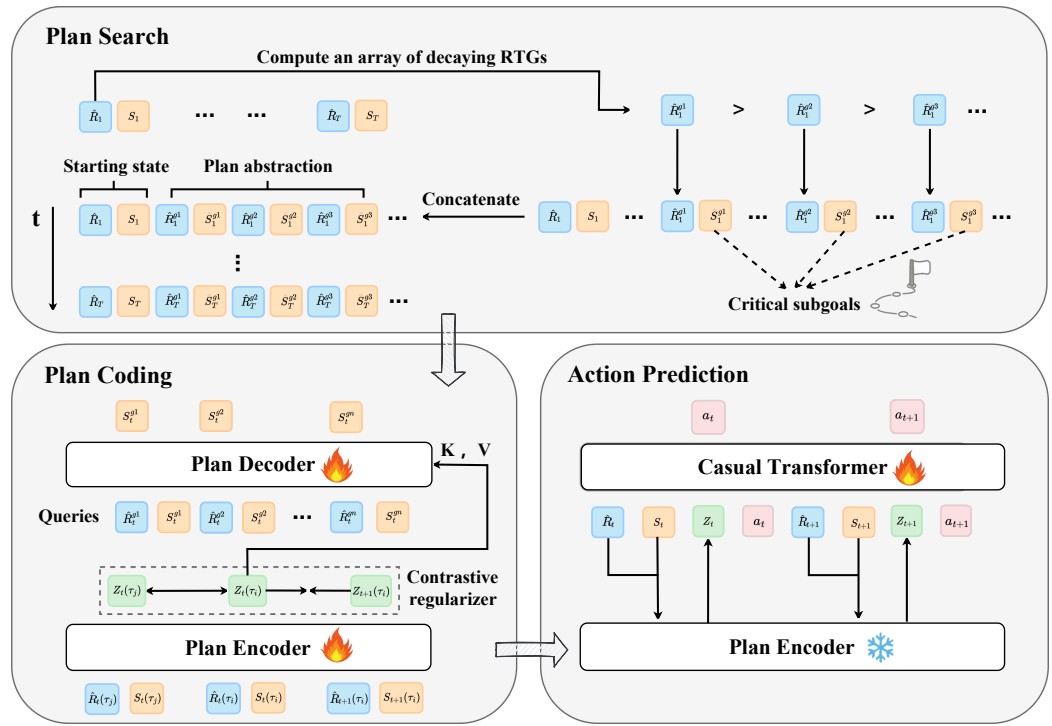

Figure 1: DT-Pro Architecture. The Plan Search module is primarily designed to select critical subgoals by leveraging decreasing RTG values in the trajectory. These subgoals are then used to train the Plan Encoder to generate high-level plan codings, which are subsequently integrated into the Decision Transformer (DT) for autoregressive action generation.

state $S_{t'}$ is selected as the $i$-th subgoal $S_t^{gi}$ for timestep $t$. After obtaining $n$ RTG-subgoal pairs $\zeta_t = (\hat{R}_t^{g1}, S_t^{g1}, \ldots, \hat{R}_t^{gn}, S_t^{gn})$, we concatenate them with the original pair $(\hat{R}_t, S_t, a_t)$ to form a subgoal-augmented representation. This operation is repeated for every timestep in the trajectory, resulting in a set of augmented trajectories $\tau' = \{\hat{R}_t, S_t, \zeta_t, a_t\}_{t=0}^{T}$, where each tuple $(\hat{R}_t, S_t, \zeta_t, a_t)$ represents the current information with temporal abstraction of the future. The details of the Plan Search are provided in Algorithm 1.

---

**Algorithm 1** Searching Critical Subgoals Based on RTG Decay

**Input**: Trajectory $\tau$, number of subgoals $n$, decay factors set $\lambda$
**Output**: Subgoal-augmented trajectories $\tau'$

1: $\lambda = \left\{ \lambda_i = \frac{n-i}{n} \mid i = 1, 2, ..., n \right\}$
2: **for** each timestep $t$ in $\tau$ **do**
3:     Initialize RTGs-subgoals $\zeta_t \leftarrow []$
4:     **for** each $\lambda_i \in \lambda$ **do**
5:         Compute target RTG: $\hat{R}_t^{gi} \leftarrow \lambda_i \cdot \hat{R}_t$, Find $t' = \arg\min_{j>t} |\hat{R}_j - \hat{R}_t^{gi}|$
6:         $\hat{R}_t^{gi} = \hat{R}_{t'}$, $S_t^{gi} = S_{t'}$, Append $\{\hat{R}_t^{gi}, S_t^{gi}\}$ to $\zeta_t$
7:     **end for**
8:     $(\hat{R}_t, S_t, \zeta_t, a_t)$ is stored in $\tau'$
9: **end for**

---

The motivation behind this design lies in the fact that $\hat{R}_t$ represents the cumulative return from the current state $S_t$ to the terminal state $S_T$. A smaller $\hat{R}_t$ indicates that the corresponding state is closer to the final goal of the task. By applying a set of decayed return targets, we retrieve a sequence of future states that approximately span the remainder of the trajectory. These subgoals serve as

milestones that effectively capture the underlying plan. Compared to existing approaches (e.g., ADT Ma et al. (2023) and WT Badrinath et al. (2023a)) that select subgoals based on naive strategies, DT-Pro provides an explainable and effective strategy for selecting critical subgoals. Our empirical studies demonstrate that the plan formed by these critical subgoals are much more informative and robust than that selected by naive strategies.

## 4.2 PLAN CODING: LEARNING A STRUCTURED PLAN SPACE

Although the critical subgoals extracted by the Plan Search module provide useful temporal abstraction of the plan, directly conditioning the policy on the subgoals will cause significant computational overload and instability due to the noisy and irrelevant information.

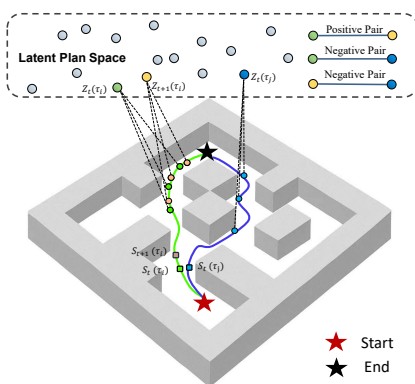

To address this, we introduce a Plan Coding module that transforms subgoal sequences into a compact latent plan. Specifically, we use a multi-layer perceptron (MLP) $f_\theta$ to encode each state $S_t$ and its associated return-to-go $\hat{R}_t$ into a latent vector $z_t = f_\theta(S_t, \hat{R}_t)$. To ensure that $z_t$ captures the essential planning structure, we train it through a Transformer-based decoder $g_\phi$ with a cross attention mechanism. Since the latent plan $z_t$ lacks direct supervision signals, we train the plan encoder and decoder in a self-supervised way by reconstructing the subgoal sequences. As is illustrated in Figure 1, each $z_t$ is paired with a sequence $\hat{S}_t^g = \{\hat{R}_t^{g1}, S_t^{g1}, \ldots, \hat{R}_t^{gn}, S_t^{gn}\}$, and used as the key and value in the decoder, while the autoregressive subgoal queries serve as input. The decoder then autoregressively predicts each subgoal token as:

Figure 2: Illustration on the latent plan space and positive/negative pairs in contrastive learning.

$$S_t^{gi'} = g_\phi(Q_t = [\hat{R}_t^{gi}, \zeta_t^{i-1}], K_t = z_t, V_t = z_t) \tag{1}$$

where $\zeta_t^{i-1} = (\hat{R}_t^{g1}, S_t^{g1}, ..., \hat{R}_t^{gi-1}, S_t^{gi-1})$. We apply a MSE loss over predicted subgoals:

$$\mathcal{L}_{\text{Pre}} = -\sum_{t=0}^{T}\sum_{i=0}^{n} || S_t^{gi} - g_\phi(\hat{R}_t^{gi}, \zeta_t^{i-1}, z_t) ||^2 \tag{2}$$

**Stabilizing plan representations via contrastive regularizer.**

The MLP-based plan encoder $f_\theta$ learns a mapping from the current state and RTG to a latent plan space. However, the learned plan space may not be stable due to the noise in the dataset. To address this, we introduce a simple yet effective contrastive regularizer He et al. (2020) to stabilize the plan representations. The idea is that the optimal plans usually do not deviate too much in adjacent time steps, while they can deviate in different trajectories. As illustrated in Figure 2, latent plans from adjacent time steps within the same trajectory are treated as positive pairs, while plans from different trajectories are treated as negatives. The contrastive loss encourages temporally consistent and trajectory-aware representations:

$$\mathcal{L}_{\text{Reg}} = -\sum_{i=1}^{N}\log\frac{\exp\big(Sim(Z_t(\tau_i), Z_{t+1}(\tau_i))\big)}{\sum_{j=1,\,j\neq i}^{N}\exp\big(Sim(Z_t(\tau_i), Z_{t'}(\tau_j))\big) * Temperature}. \tag{3}$$

The final loss for the Plan Coding module can be represented by a weighted sum of two losses:

$$\mathcal{L}_{\text{PC}} = \mathcal{L}_{\text{Pre}} + \eta\mathcal{L}_{\text{Reg}}. \tag{4}$$

The details of Plan Coding are provided in Algorithm 3.

## 4.3 ACTION PREDICTION: CONDITIONING ACTION ON THE LATENT PLAN

After the pretraining at the first stage, the Plan Encoder can generate high-level plan representations by taking the current state and RTG as inputs. To integrate this high-level planning information into the action generation process, we slightly modify the architecture of the original DT by imposing an implicit planning step. Specifically, at each timestep $t$, we freeze the Plan Encoder and compute a latent plan representation $z_t = f_\theta(S_t, \hat{R}_t)$, where $f_\theta$ is the pretrained Plan Encoder. Note that we freeze the Plan Encoder during the second training stage, in order to protect the structured plan space from being distorted. Then, DT-Pro takes as input a sequence of tuples $(\hat{R}_t, S_t, z_t, a_t)$ and learns to autoregressively predict the current action $a_t$ based on the historical context $\tau_{t-k+1:t-1}$ and the current input $(\hat{R}_t, S_t, z_t)$. The training loss at the second stage can be written as:

$$L_{\text{DT-Pro}} = -\sum_{t=0}^{T} || \, a_t - \pi_\delta \left( \tau_{t-k+1:t-1}, S_t, r_t, \lfloor z_t \rfloor \right) ||^2 \tag{5}$$

where $\pi_\delta$ denotes the parameters of the DT-Pro policy network, and $\lfloor z_t \rfloor$ represents the application of a stop-gradient operator on the latent planning vector $z_t$. By conditioning action prediction on its associated plan, DT-Pro gains a strong planning capability so that it would select global optimal actions with higher probabilities. Moreover, DT-Pro reasons future plans in a compact latent space, which allows efficient replanning at each time step with only a marginal increase in the computational cost. Note that adjusting plans in real time could bring significant performance improvement, especially in long-horizon tasks.

## 4.4 OVERALL TRAINING AND DEPLOYMENT

In this section, we will summarize the overall training procedures of DT-Pro and show how it can be extended to sparse reward environments. In the first stage of training, DT-Pro first obtains subgoal information through the Plan Search module and adds it to the current trajectory. Then, subgoal-augmented trajectories are used to train the Plan Encoder, which takes state and RTG as inputs and outputs a latent plan. In the second stage of training, the Plan Encoder is frozen and a causal Transformer model is trained to predict action based on the current state, RTG and latent plan. The overall training process of DT-Pro is described in Algorithm 2.

**Extending DT-Pro to sparse reward environments.** As the plan search algorithm 1 relies on the RTGs of sequences, it does not directly apply in sparse reward environments where only the final reward of the trajectory is provided. To address this, we pretrain a DT model and use it as a critic to provide granular RTG signals for the plan search. As is shown in Section 5.5 of the original DT paper Chen et al. (2021), pretrained DT can provide reasonable RTG signals for each timestep. Appending these predicted RTGs to the original trajectories would help our lan search algorithm to effectively find critical subgoals and associated plans. The experimental results in Section 5.2.2 demonstrate that DT-Pro can be extend to sparse reward settings with only slight performance drop.

---

**Algorithm 2** Overall training process of DT-Pro

---

**Input**: Trajectory $\tau$, target number of subgoals $n$, decay factors set $\lambda$
**Output**: DT-Pro policy $\pi_\delta$

    **if** sparse reward environments **then**
        Predict RTG using the trained DT model $DT$: $\hat{R} = DT(\tau)$
        Replace the inaccurate and sparse RTG in the original data with $\hat{R}$
    **end if**
    # Input $\tau$ into Algorithm 1 to obtain subgoals and add them to trajectory $\tau$ .
    $\tau'$ = Plan Search($\tau$, n, $\lambda$)
    # Input trajectory $\tau'$ into Algorithm 3 to train the Plan Encoder (Decoder will be discarded).
    Plan Encoder $f_\theta$ = Plan Coding($\tau'$)
    #After Plan Coding training, the Encoder parameters will be frozen.
    Freeze the Plan Encoder and train DT-Pro policy $\pi_\delta$ according to Algorithm 4.

---

| Environment | CQL | IQL | DT | ADT | WT | LPT-EI | DT-Pro (Ours) |
|---|---|---|---|---|---|---|---|
| Half-M | $44.0_{\pm 5.4}$ | $\mathbf{47.4}_{\pm 0.2}$ | $42.4_{\pm 0.2}$ | $\mathbf{48.7}_{\pm 0.2}$ | $43.0_{\pm 0.2}$ | $43.5_{\pm 0.1}$ | $43.1_{\pm 0.1}$ |
| Half-MR | $\mathbf{45.5}_{\pm 0.5}$ | $44.2_{\pm 1.2}$ | $35.4_{\pm 1.6}$ | $42.8_{\pm 0.2}$ | $39.7_{\pm 0.3}$ | $40.6_{\pm 0.2}$ | $\mathbf{43.2}_{\pm 0.1}$ |
| Half-ME | $91.6_{\pm 2.8}$ | $86.7_{\pm 5.3}$ | $84.9_{\pm 1.6}$ | $91.7_{\pm 1.5}$ | $\mathbf{93.2}_{\pm 0.5}$ | $89.3_{\pm 0.2}$ | $\mathbf{93.0}_{\pm 0.6}$ |
| Hopper-M | $58.5_{\pm 2.1}$ | $66.2_{\pm 5.7}$ | $63.5_{\pm 5.2}$ | $60.6_{\pm 2.8}$ | $63.1_{\pm 0.1}$ | $63.8_{\pm 0.3}$ | $\mathbf{77.2}_{\pm 4.7}$ |
| Hopper-MR | $\mathbf{95.2}_{\pm 6.4}$ | $92.7_{\pm 8.6}$ | $43.3_{\pm 23.9}$ | $83.5_{\pm 9.5}$ | $88.9_{\pm 2.4}$ | $89.9_{\pm 0.6}$ | $\mathbf{92.9}_{\pm 2.2}$ |
| Hopper-ME | $105.4_{\pm 6.8}$ | $91.5_{\pm 14.3}$ | $100.6_{\pm 8.3}$ | $101.6_{\pm 5.4}$ | $\mathbf{110.9}_{\pm 0.6}$ | $109.8_{\pm 2.1}$ | $\mathbf{111.7}_{\pm 0.5}$ |
| Walker2d-M | $72.5_{\pm 0.8}$ | $78.3_{\pm 8.7}$ | $69.2_{\pm 4.9}$ | $80.9_{\pm 3.5}$ | $74.8_{\pm 1.0}$ | $81.2_{\pm 0.3}$ | $\mathbf{81.4}_{\pm 0.2}$ |
| Walker2d-MR | $\mathbf{77.2}_{\pm 5.5}$ | $73.8_{\pm 7.1}$ | $58.9_{\pm 7.1}$ | $\mathbf{86.3}_{\pm 1.1}$ | $67.9_{\pm 3.4}$ | $75.7_{\pm 0.4}$ | $76.0_{\pm 1.9}$ |
| Walker2d-ME | $108.8_{\pm 0.7}$ | $109.6_{\pm 1.0}$ | $89.6_{\pm 38.4}$ | $\mathbf{112.1}_{\pm 0.4}$ | $109.6_{\pm 1.0}$ | $108.6_{\pm 2.3}$ | $\mathbf{109.7}_{\pm 1.0}$ |
| Maze2d-UD | $14.4$ | $57.8_{\pm 12.5}$ | $-6.8_{\pm 10.9}$ | $51.6_{\pm 8.1}$ | $51.6_{\pm 20.4}$ | $\mathbf{70.6}_{\pm 1.4}$ | $69.0_{\pm 4.6}$ |
| Maze2d-MD | $30.5$ | $28.1_{\pm 16.8}$ | $31.5_{\pm 3.7}$ | $50.6_{\pm 15.9}$ | $\mathbf{65.8}_{\pm 21.5}$ | $46.7_{\pm 0.8}$ | $\mathbf{98.0}_{\pm 3.2}$ |
| Average | $67.6$ | $70.8$ | $69.0$ | $73.7$ | $73.5$ | $\mathbf{74.5}$ | $\mathbf{81.4}$ |

Table 1: Performance comparisons on D4RL and Maze2d benchmarks. For each environment, the top-2 performance results are highlighted in **bold**.

## 5 EXPERIMENTS

In this section, we conducted comprehensive experiments to evaluate DT-Pro from three perspectives: overall performance, module effectiveness, and hyperparameter sensitivity Specifically, we first evaluate DT-Pro against a set of strong baselines on benchmark tasks to validate its overall effectiveness. We then conduct ablation studies to assess the contribution of key components. Finally, to better understand the design choices, we further analyze the effect of the number of subgoals on performance. In additional, we provide computational cost analysis, robustness evaluation and visualization of decoded plans in Appendix D.

### 5.1 EXPERIMENTAL SETUP

We evaluate DT-Pro on the widely-used D4RL benchmark suite, which covers a diverse set of offline reinforcement learning tasks. In our experiments, we focus on two representative task families that reflect different types of decision-making challenges: **Gym-MuJoCo**, **Maze2d**. The detailed experimental settings are described in Appendix C.To ensure robustness and statistical significance, all experiments are run with 4 random seeds. We report the average return and standard deviation across these seeds.

### 5.2 PERFORMANCE COMPARISONS WITH PRIOR WORKS

#### 5.2.1 RESULTS IN DENSE REWARD ENVIRONMENTS

To evaluate the effectiveness of DT-Pro, we compare its performance with a diverse set of representative baselines: CQL Kumar et al. (2020), IQL Kostrikov et al. (2021), DT Chen et al. (2021), Autotuned Decision Transformer(ADT) Ma et al. (2023), Waypoint Transformer (WT) Badrinath et al. (2023a) and Latent Plan Transformer(LPI-EI) Kong et al. (2024). Detailed descriptions of the environments and baselines are given in Appendix C.

**Analysis.** Table 1 summarizes the normalized performance of DT-Pro across all evaluated environments. Our method achieves an average score of 81.4, substantially outperforming the previous best baseline ADT (73.7) and establishing a new state-of-the-art on the D4RL benchmark. In the Gym-MuJoCo tasks, DT-Pro consistently outperforms existing methods across all dataset settings. Notably, on the challenging medium-replay datasets—where data quality is lower and trajectories are more diverse—DT-Pro achieves substantial gains over DT, with average improvements of over 50 points in HalfCheetah, Hopper, and Walker2d. These results demonstrate the robustness of DT-Pro under suboptimal data distributions. In the Maze2d navigation benchmarks, which emphasize long-horizon, goal-conditioned decision making, DT-Pro also achieves strong performance. On the medium-dense variant, DT-Pro improves upon the best-performing prior method (WT) by 33.6%, and maintains leading performance across all difficulty levels. These results show that DT-Pro effectively enhances the planning ability of sequence modeling methods in long-horizon tasks.

### 5.2.2 RESULTS IN SPARSE REWARD ENVIRONMENTS

To evaluate the effectiveness of our algorithm on sparse reward datasets, we use the trained DT model as the critic and input the sparse reward dataset into the DT model to predict more RTG information (According to 4.4). Then, combining with our method, we obtain effective subgoals. To test the validity of the subgoals, we conduct experiments on a delayed return version of the Hopper environment, with the results shown in the Table 2. When CQL's performance significantly drops in sparse environments, DT-pro is less affected, demonstrating that DT-pro can still achieve relatively good results under sparse reward conditions.

| Dataset | Delayed (Sparse) | | | Original (Dense) | | |
|---|---|---|---|---|---|---|
| | **DT-pro (Ours)** | **CQL** | **ADT** | **DT-pro (Ours)** | **CQL** | **ADT** |
| Hopper-ME | **104.6 $\pm$ 2.5** | $9.0 \pm 1.3$ | $100.9 \pm 2.3$ | **111.7 $\pm$ 0.5** | $105.6 \pm 6.8$ | $101.6 \pm 5.4$ |
| Hopper-M | **73.2 $\pm$ 3.8** | $5.2 \pm 1.3$ | $57.9 \pm 4.3$ | **77.2 $\pm$ 4.7** | $58.5 \pm 2.1$ | $60.6 \pm 2.8$ |
| Hopper-MR | **85.3 $\pm$ 2.6** | $2.0 \pm 1.9$ | $80.6 \pm 3.8$ | $92.9 \pm 2.2$ | **95.2 $\pm$ 6.4** | $83.5 \pm 9.5$ |

Table 2: Standard score on a delayed return version of the Hopper environment.

### 5.3 ABLATION STUDIES

#### 5.3.1 EFFECTIVENESS OF THE PLAN SEARCH

In Section 4.1, we introduce a data preprocessing pipeline to extract informative subgoals, which are later used to guide the planning process. The quality of these extracted subgoals is critical, as it directly affects the Planning Generator's ability to produce meaningful plans and thus influences the overall decision-making performance of DT-Pro. To evaluate the importance of subgoal quality, we compare our approach against two baseline extraction strategies:*Random subgoals*: At each timestep, $n$ subgoals are randomly sampled from the same trajectory. This naive approach ignores the relevance between subgoals and the current decision-making context. *Fixed-interval subgoals*: Subgoals are selected at fixed temporal intervals (e.g., $t = 20, 40, 60, 80, 100$ for a length-100 trajectory). This strategy offers better stability than random selection but still lacks adaptiveness. It overlooks the actual task structure and the varying importance of different trajectory segments.

As shown in Figure 3, random subgoals often lead to significantly worse results with high variances, particularly on datasets like `hopper-medium`, where the quality of behavior data is already limited. This is because random subgoals often consist of uninformative or misleading states. For fixed-interval subgoals, while this approach achieves slightly more consistent results, its performance remains inferior to DT-Pro with plan search, due to its failure to identify critical subgoals that carry more useful information.

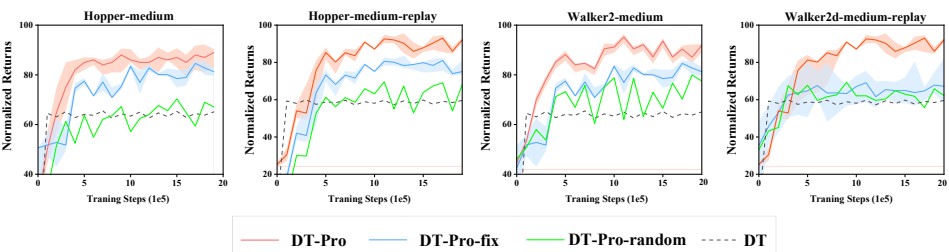

Figure 3: Learning curves of different subgoal selection strategies, where DT-Pro with the plan search shows a clear advantage.

#### 5.3.2 EFFECTIVENESS OF THE CONTRASTIVE REGULARIZER

To improve the quality of plan representations and mitigate the influence of noisy or ambiguous subgoal sequences, we introduce a contrastive learning objective into the Plan Coding module. To assess its contribution, we conduct an ablation study by removing the contrastive loss from the training objective. As illustrated in Figure 4, while the performance gap in Walker2d is marginal, contrastive learning shows clear advantages in Maze2d tasks. This discrepancy is attributable to the nature of the environments: Maze2d involves complex, goal-directed trajectories with diverse subgoal structures,

which benefit from the structured representations encouraged by contrastive learning. In contrast, Walker2d primarily involves simple, linear motion with limited hierarchical planning, rendering its benefits from the plan coding as well as the contrastive regularizer. However, in many real-world complex problems, a compact and structured latent plan space would greatly improve the quality.

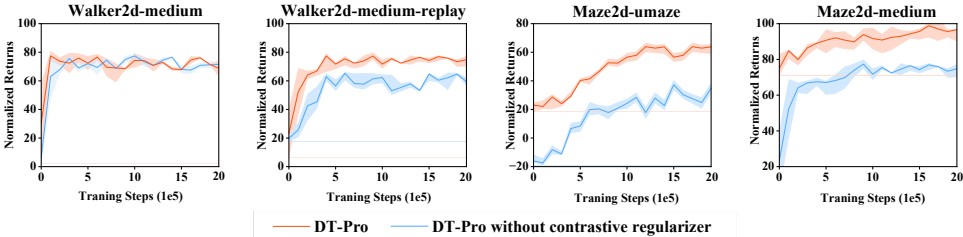

Figure 4: Learning curves of DT-Pro and DT-Pro without contrastive regularizer.

## 5.4 PERFORMANCE ON DIFFERENT NUMBERS OF SUBGOALS

As described in the Plan Coding module, the latent plan representation is guided by multi-step subgoals extracted from trajectories, where the number of subgoals represents different levels of plan abstraction. We now investigate how the number of subgoals $N$ influences the abstraction level of the generated plan. As shown in Figure 5, increasing the number of subgoals initially improves performance in both Walker2d and Maze2d, as a richer set of subgoals offers better temporal abstraction and helps the planner capture future structure more effectively. With a moderate number, each subgoal often reflects a meaningful intermediate milestone, enhancing the interpretability and utility of the plan. However, beyond a certain point, further

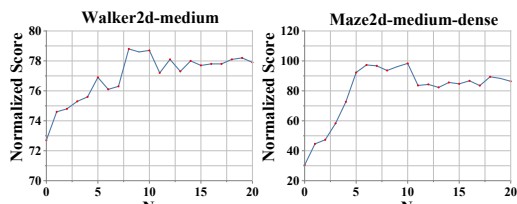

Figure 5: Standardized scores of DT-Pro under different numbers of subgoals on the Walker2d-medium dataset (left) and the Maze2d-medium-dense dataset (right).

increasing $N$ results in performance degradation. This is primarily due to the redundancy introduced by overly dense subgoal sequences, which can obscure the core trajectory structure and inject noise into the plan representation. Furthermore, additional subgoals impose a greater prediction burden, and under constrained capacity or noisy data, may accumulate errors that compromise the integrity of the plan. These results present a trade-off: while temporal abstraction facilitates long-term reasoning, overly fine-grained decomposition can harm the plan quality. In addition, the optimal choice of $N$ may vary in different environments, which reveals that different environments require different levels of plan abstraction. This aligns with our intuition that strategic environments may require more subgoals to form effective plans.

## 6 CONCLUSION

Decision Transformer represents a straightforward and effective method for offline reinforcement learning. Nevertheless, existing methods often result in myopic policies. In this study, we introduce DT-Pro, a novel framework for sequence modeling aimed at augmenting DT with a simple yet powerful mechanism for implicit planning. By integrating a plan search module to identify critical subgoals, a plan coding module to create compact latent plan representations, and an action prediction module to generate farsighted actions, DT-Pro addresses key limitations of traditional DT in long-horizon tasks. Notably, DT-Pro overcomes challenges such as myopic decision-making and computational inefficiency by reasoning plans in a structured plan space. Experiments across diverse control and navigation benchmarks demonstrate that DT-Pro significantly outperforms both offline reinforcement learning methods and existing DT variants. Moreover, benefiting from the latent space planning, DT-Pro maintains a strong performance against sparse reward and noisy states.

ETHICS STATEMENT:

This work does not involve human subjects, animals, or sensitive data. All authors adhere to the ICLR Code of Ethics and have no conflicts of interest to disclose. The research was conducted following all applicable legal and ethical standards. No ethical concerns were raised during the development or submission of this paper.

REPRODUCIBILITY STATEMENT:

The entire process of this work is fully reproducible. The datasets used in this study are sourced from the D4RL dataset, and the relevant code for downloading the datasets will be provided in the supplementary materials. The detailed steps of the experiments will also be outlined in the supplementary materials. Additionally, the source code for the model will be available in the supplementary materials to enable reproduction of all the experimental results.

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

## A  LLM USAGE DISCLOSURE

In the course of this research, the authors did not use any Large Language Models (LLMs) as tools for research ideation, writing, or data analysis. All content of this study has been independently created by the authors, without reliance on any form of artificial intelligence language models for research work or content generation during the writing process.

## B  SUPPLEMENTARY ALGORITHMS

Algorithm 3 mainly explains Plan Coding: A Plan Encoder is trained on the Subgoals obtained from Algorithm 1, and after training, the Encoder parameters are frozen.

---
**Algorithm 3** Contrastive Regularized Plan Coding

---
1: # **Input:** A set of trajectories $\tau = \{\tau_1, \tau_2, \ldots, \tau_N\}$
2: # **Output:** Plan Encoder $f_\theta$
3: **for** each trajectory $\tau \in \{\tau_1, \tau_2, \ldots, \tau_N\}$ **do**
4:    **for** each timestep $t$ in $\tau$ **do**
5:       Compute latent plan: $z_t = \text{Encoder}(S_t, \hat{R}_t)$
6:       # Train decoder in an auto-regressive manner to predict subgoal sequence $\zeta'_t$
7:       $\zeta'_t = \text{Decoder}(Q_t = \zeta_t, K_t = z_t, V_t = z_t)$
8:       Obtain the subgoal prediction loss: $\mathcal{L}_{\text{Pre}} = \|\zeta_t - \zeta'_t\|^2$
9:    **end for**
10: **end for**
11: **for** each trajectory $\tau_i$ in all trajectories **do**
12:    **for** each timestep $t$ in trajectory $\tau_i$ **do**
13:       # Positive sample similarity:
14:       $positive\_similarity \leftarrow \text{Sim}(Z_t(\tau_i), Z_{t+1}(\tau_i))$
15:       # Negative sample similarity sum:
16:       **for** each trajectory $\tau_j$ in all trajectories, where $j \neq i$ **do**
17:          $negative\_sum + = \text{Sim}(Z_t(\tau_i), Z_t(\tau_j))$
18:       **end for**
       $\mathcal{L}_{\text{Reg}} + = -\log\left(\frac{positive\_similarity}{positive\_similarity + negative\_sum}\right)$
19:    **end for**
20: **end for**
21: Minimize the final encoder loss: $\mathcal{L}_{\text{encoder}} = \mathcal{L}_{\text{Pre}} + \eta \mathcal{L}_{\text{Reg}}$

---

Algorithm 4 mainly explains the Action Prediction module. The Encoder, trained in Algorithm 3, can be embedded into the DT model to assist in planning and achieve better performance.

---
**Algorithm 4** Training the policy of DT-Pro

---
1: # **Input:** A set of trajectories $\tau = \{\tau_1, \tau_2, \ldots, \tau_N\}$
2: # **Output:** Predicted actions for each timestep
3: **for** each trajectory $\tau \in \{\tau_1, \tau_2, \ldots, \tau_N\}$ **do**
4:    **for** each timestep $t$ in $\tau$ **do**
5:       Compute latent plan: $z_t = \text{Encoder}(S_t, \hat{R}_t)$
6:       Predict action: $a'_t = \text{DT-Pro}(\hat{R}_t, S_t, z_t)$
7:       Minimize the action prediction loss: $\text{L}_{\text{DT-pro}} = \|a'_t - a_t\|^2$
8:    **end for**
9: **end for**

---

## C    EXPERIMENTAL DETAILS

### C.1    ENVIROMENT DESCRIPTIONS

**Gym-MuJoCo:** This suite includes classic continuous control tasks simulated in the MuJoCo physics engine. We select three standard environments—Hopper, Walker2d, and HalfCheetah—each characterized by high-dimensional continuous state and action spaces. The objective in these environments is to learn locomotion policies that produce stable and efficient forward movement. For each environment, D4RL provides multiple datasets of varying quality, including medium, medium-replay, and medium-expert, enabling evaluation under different data regimes.

**Maze2d:** This set of tasks emphasizes long-horizon, goal-conditioned navigation. The agent must navigate a 2D maze with obstacles to reach a specified goal location from a fixed start position. These tasks require strong planning capabilities and are well-suited for evaluating the benefit of future-aware modeling. We use three versions with increasing difficulty: umaze-dense, medium-dense, and large-dense, all of which provide dense rewards for smoother learning.

### C.2    BASELINES ALGORITHMS

**Offline reinforcement learning**: including CQL Kumar et al. (2020), and IQL Kostrikov et al. (2021), which are value-based or actor-critic based approaches designed for offline RL settings.

**Sequence modeling with Transformers**: including DT Chen et al. (2021), Autotuned Decision Transformer(ADT) Ma et al. (2023), Waypoint Transformer (WT) Badrinath et al. (2023a) and Latent Plan Transformer(LPI-EI) Kong et al. (2024)which formulate policy learning as sequence modeling with future conditioning.

All experimental data comes from Waypoint DT, ADT, and LPT, except for the LPTKong et al. (2024) data in the Walker-medium-expert, Hopper-medium-expert, and HalfCheetah-medium-expert datasets. For these three datasets, we used the code provided in the LPT paper to test and obtain the corresponding scores.

### C.3    RETURN TARGETS USED IN EVALUATION

The target returns for the Gym-MuJoCo and Maze2d tasks, presented in Table 3, are given as normalized scores. For Gym-MuJoCo, most target returns are adopted from prior work such as DT, while for Maze2d, we primarily use the highest trajectory scores available in the existing datasets, as provided on the official D4RL website.

Table 3: DT-Pro target return settings across Gym-MuJoCo and Maze2d benchmarks

| Environment | Target Returns |
| --- | --- |
| Hopper-Medium | 3600 |
| Walker2d-Medium | 4000 |
| Halfcheetah-Medium | 6000 |
| Hopper-Medium-Replay | 3600 |
| Walker2d-Medium-Replay | 4000 |
| Halfcheetah-Medium-Replay | 9000 |
| Hopper-Medium-Expert | 3600 |
| Walker2d-Medium-Expert | 5000 |
| Halfcheetah-Medium-Expert | 12000 |
| Maze2d-umaze-dense | 100 |
| Maze2d-medium-dense | 300 |

## C.4 HYPER-PARAMETERS

The hyperparameters used by DT-pro in our experiments are summarized in Table 4. Most of them follow the default configuration of the DT. In addition, we incorporate a contrastive regularizer during training, with the temperature coefficient set to 0.05, 0.1, and 0.2. The number of subgoals varies across different datasets, primarily because the trajectory lengths differ significantly across environments. As a result, the number of subgoals that can be meaningfully extracted also varies. Nevertheless, we typically select between 5 and 10 subgoals.

The hyperparameters used for the encoder and decoder of the plan module are listed in Table 5. To ensure that the learned plan representation aligns well with the underlying state space, we adapt the output dimensionality of the plan according to the state dimensionality of different environments. Specifically, we set the plan dimension to 16 for Gym-MuJoCo environments and 4 for Maze2d environments, reflecting the fact that the typical state dimension is around 16 in Gym-MuJoCo and 4 in Maze2d.

This adaptive design allows the plan embedding to retain sufficient representational capacity without introducing unnecessary complexity. Overall, our Plan framework remains lightweight in terms of parameter count and computational overhead, while still demonstrating strong empirical performance across benchmarks.

Table 4: DT-pro Hyper-parameters

|  | **Hyper-parameter** | **Value** |
| --- | --- | --- |
| **Architecture** | Hidden layers | 4 |
|  | Hidden dim | 128 |
|  | Heads num | 4 |
|  | Clip grad | 0.25 |
|  | Embedding dim | 128 |
|  | Embedding dropout | 0.1 |
|  | Activation function | ReLU |
|  | Sequence length | 20 |
| **Training** | Optimizer | AdamW |
|  | Learning rate | 1e-4 |
|  | Batch size | 256 |
|  | Weight decay | 1e-4 |
|  | Warm-up steps | 1e5 |

Table 5: Transformer-based Plan Decoder Hyper-parameters

|  | **Hyper-parameter** | **Value** |
| --- | --- | --- |
| **Architecture** | Hidden layers | 2 |
|  | Heads num | 16 |
|  | Clip grad | 0.25 |
|  | Embedding dim | 128 |
|  | Embedding dropout | 0.1 |
|  | Activation function | ReLU |
| **Training** | Optimizer | AdamW |
|  | Learning rate | 1e-4 |
|  | Batch size | 128 |

# D  ADDITIONAL EXPERIMENTS

## D.1  COMPUTATIONAL COST

All experiments were conducted on NVIDIA Tesla P100-PCIE GPUs. The computational cost of our framework is distributed across three primary modules. The first module, Plan Search, operates on small-scale benchmark datasets and runs entirely offline before training, incurring negligible cost. The second module, Plan Coding, includes a light-weight MLP-based encoder and a Transformer-based decoder. Given that the number of subgoals is typically capped at 20, the sequence length remains manageable, and this stage adds only 8%–12% to the total training time of the DT. The third module, Action Prediction, extends the standard DT with an additional MLP to encode the plan. Importantly, this modification preserves the core architecture and hyperparameter settings (see Appendix A for details), resulting in only a marginal increase of 10%–12% in training time. Despite this small overhead, as shown in Table 1, DT-Pro achieves a substantial performance improvement of approximately 42% over the vanilla DT on benchmark tasks. This demonstrates that incorporating plan encoding delivers significant gains with a relatively low computational cost.

## D.2  PERFORMANCE ON DIFFERENT NUMBERS OF SUBGOALS(APPENDIX)

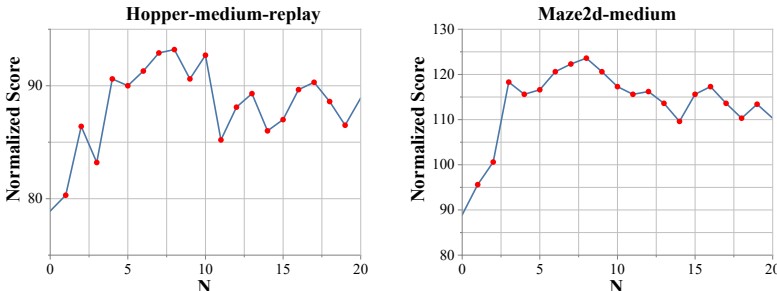

Figure 6: Standardized scores of DT-Pro under different numbers of subgoals on the Walker2d-medium dataset (left) and the Maze2d-medium-dense dataset (right).

## D.3  EVALUATING THE ROBUSTNESS OF PLAN

Methods like ADT rely on manually preset hyperparameters to select a specific state in the trajectory as a subgoal. However, this approach does not guarantee the quality of the subgoals. In contrast, our algorithm relies on multiple subgoals during the reasoning process of the plan. Therefore, the question arises: Is the plan overly sensitive to the quality of certain subgoals? The experimental setup is as follows: Gaussian noise is randomly added to the extracted subgoals, with noise levels of 20%, 40%, and 60%. The experimental results are shown in the Fig 7. The addition of Gaussian noise to the subgoal in ADT results in a significant decline. In contrast, when a small number of subgoals have poor quality, the impact on the algorithm's performance is negligible. As the noise proportion increases, the algorithm's performance does decline, but it remains at a relatively high level. Even when more than half of the subgoal data is of low quality, the algorithm's performance is still comparable to that of ADT. These results validate that the Plan method demonstrates robust performance when dealing with imperfect data.

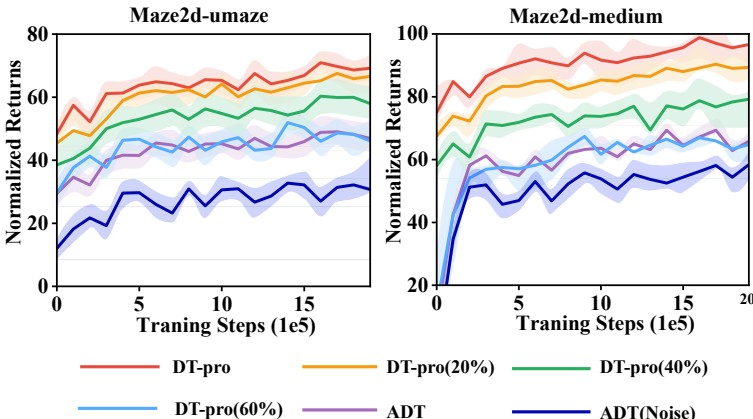

Figure 7: Learning curves regarding the addition of Gaussian noise at different proportions.

## D.4 VISUALIZATION OF DECODED PLANS

Since the plan is encoded as a high-dimensional latent vector, it cannot be directly visualized, making its interpretability and effectiveness difficult to assess. To address this, we leverage the fact that the plan serves as the key and value in the Plan Decoder to autoregressively generate subgoals, we visualize the output subgoal sequences produced by the trained decoder conditioned on the plan. This allows us to assess whether the latent plan encodes meaningful structure and intent. As shown in Figure 8 and Figure 9, we select representative states in the Maze2d: medium environment, Large environment and visualize their corresponding decoded subgoals. The visualization results show that the latent plan consistently produces goal-directed subgoal trajectories, demonstrating that the plan encoder captures informative planning signals that can effectively guide the action prediction.

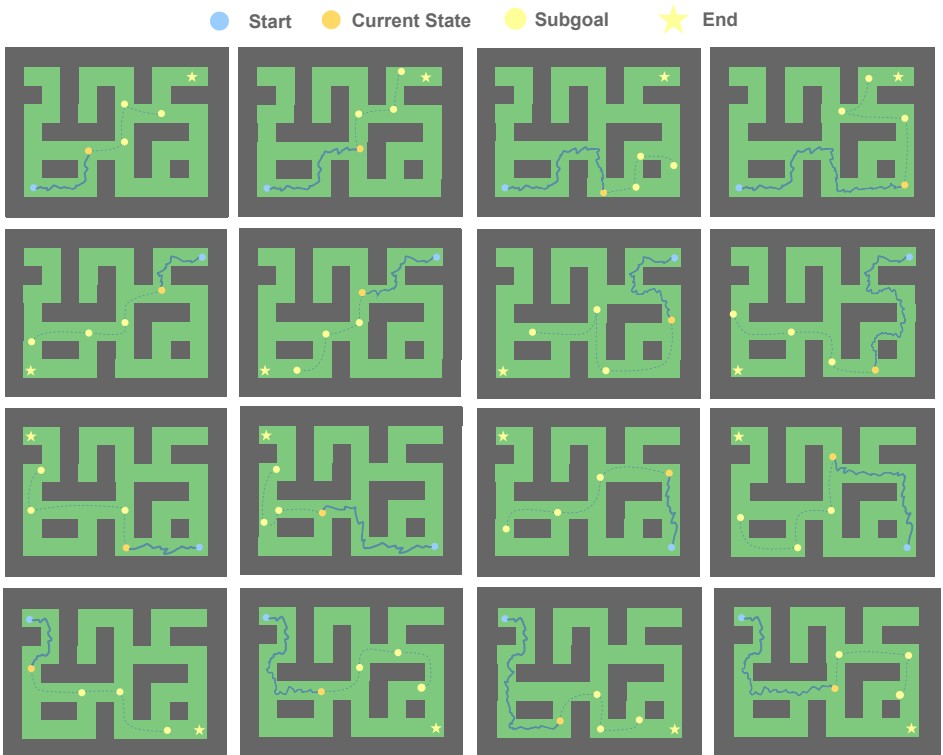

Figure 8: Visualization of subgoals decoded from latent plans of consecutive states during testing in the Maze2d-Large environment.

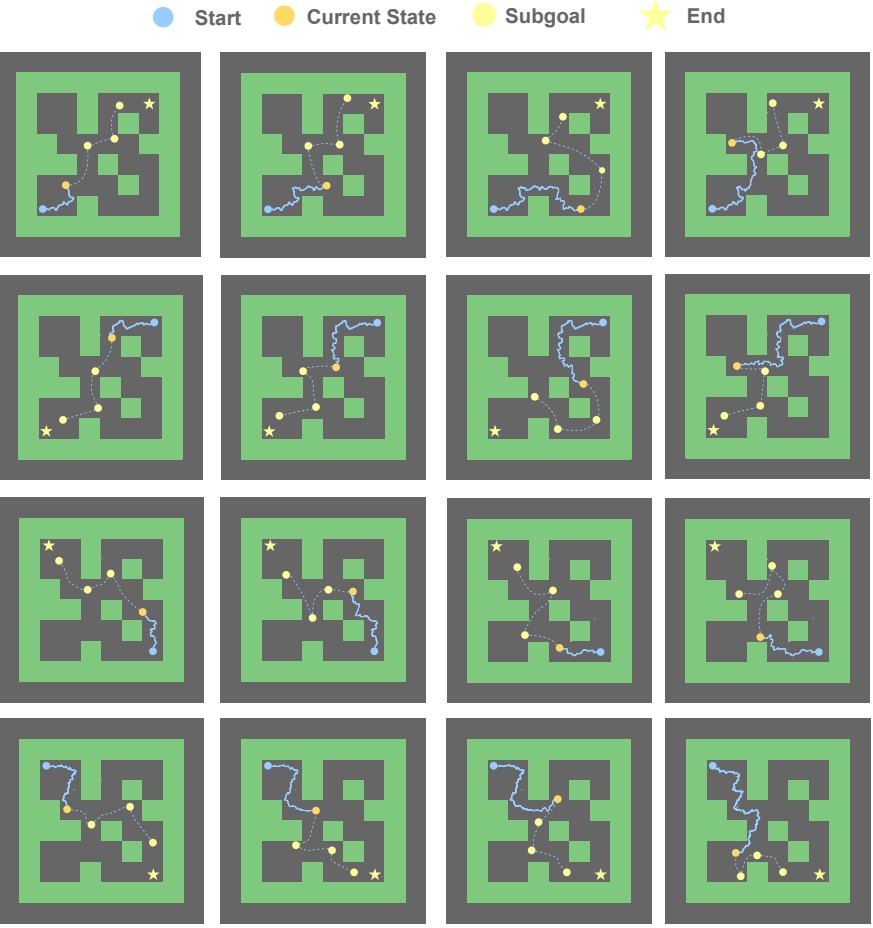

Figure 9: Visualization of subgoals decoded from latent plans of consecutive states during testing in the Maze2d-medium environment. The decoded subgoals always indicate the optimal path to the end point, demonstrating the effectiveness of the plan encoder.

