# OpenReview forum: "DT-Pro: Proactive Decision Transformers with Implicit Latent Space Planning"
_ICLR.cc/2026/Conference — Submitted to ICLR 2026_

### Official Review · Reviewer_32bG · 2025-10-16

**Soundness:** 2
**Presentation:** 3
**Contribution:** 3
**Rating:** 4
**Confidence:** 4

**Summary:**

The paper proposes DT-Pro, a Decision-Transformer variant that inserts an implicit latent-plan step.
Pipeline:
(i) mine "critical" subgoals per trajectory using a decayed-RTG heuristic (Alg. 1),
(ii) encode the subgoal sequence into a compact latent with a contrastive regularizer, and
(iii) condition a DT-style action model on that latent (encoder frozen in stage-2) with step-wise replanning at test time.
The paper reports a higher average normalized return than DT/ADT/WT/CQL/IQL on D4RL Gym-MuJoCo and Maze2d, shows ablations (subgoal strategy, contrastive term, #subgoals), a sparse-reward variant using a DT-predicted RTG signal, and provides decoded-plan visualizations.

**Strengths:**

- Simple, targeted idea: Implicit latent plan sidesteps brittle explicit waypoints, integrates cleanly with DT.
- Empirical signal on long horizons: Consistent gains on D4RL Gym-MuJoCo and Maze2d, subgoal-mining ablation is convincing.
- Interpretability: Decoded subgoal visualizations suggest some learned temporal structure.
- Scope & ablations: Vary #subgoals and (claimed) contrastive regularization, explore a sparse-reward setting.
- Potential impact: If baselines are re-established under a common protocol with stronger statistics, DT-Pro could be a go-to DT variant for longer-horizon offline RL.

**Weaknesses:**

- Objective inconsistency (paper vs. appendix vs. code): Main text describes CE/log-likelihood (plus contrastive) for subgoal decoding/policy, whereas algorithms use squared-error. The supplementary code appears to optimize MSE at both stages, with no contrastive term. This must be reconciled and reflected in the results.
- Baseline provenance/fairness: Many baseline scores are imported (only a subset rerun), risking potential protocol drift (normalization/eval differences). Stronger conclusions require rerunning baselines under a unified pipeline.
- Evaluation protocol clarity: Missing or underspecified train/val/test splits, checkpoint selection, exact D4RL normalization, and number of eval rollouts.
- "Optional plan decoding" ambiguity: Abstract hints at optional decoding/search, but evaluation appears not to use any test-time plan decoding, clarify and, if applicable, report results with/without it.
- Minor polish: Fix ADT average (73.7 vs. text 74.9), make captions self-contained (units, seeds), and add raw returns in the appendix.
- Reproducibility: Supplementary code indicates default training uses MSE objectives and omits contrastive loss. Evaluation relies on encoder outputs, with no plan-sequence decoding at test time.

**Questions:**

1. Losses: Are subgoal decoding and policy trained with CE/log-likelihood (main text) or L2 (Appendix Algs 3–4)? What does the released code actually optimize? If both were tried, please report a comparison.
2. Baseline protocol: For Table 1, which baselines were re-run under your pipeline and which were imported? How did you ensure identical target returns, normalization, number of eval episodes, and scoring for imported numbers?
3. Evaluation splits: What are your train/val/test and checkpoint-selection rules in offline replay? Please state D4RL normalization and eval rollout counts precisely.
4. Capacity/budgets: Sensitivity to plan dimension, N subgoals (partially reported), and any plan-search budget. A small grid on Hopper-MR and Maze2d-medium would help.
5. Sparse-reward: Using a pretrained DT to densify RTG - did you check for train/test contamination? How sensitive are results to the critic’s quality? Please add stds for all methods in Table 2.
6. Test-time decoding/search: The abstract implies an optional decoding/search, sections.4.3/4.4 suggests it’s not needed. Did you try multiple latent-plan samples or a beam over subgoals at inference? If yes, report it. If not, clarify.

---

> ### Author Response · Authors · 2025-11-19
> **Reply Rebuttal Comment by Authors**
>
> Thank you for your insightful feedback
>
> **Q1: Concerns of Loss Function**
>
> This was an oversight in our work. The actual loss function implemented in our experiments is **Mean Squared Error (MSE)**. The formula presented in the paper corresponds to the initial design intended for discrete environments during the early stages of the project, and we will promptly correct this discrepancy in the revised manuscript.
>
> **Q2: Concerns of Baseline Protocol**
>
> We directly adopted the experimental data of CQL and IQL, while reproducing DT, ADT, WT, and LPT-EI using our own hyperparameters. Experimental results showed that the performance of these reproduced models was mostly **inferior** to the results reported in their original papers. To ensure the strictness and fairness of the baselines, we ultimately chose to use the superior public data from the original studies.
>
> During the reproduction process, we referenced the core running logic of **DT**: fixed target returns were used in each evaluation (eval) step, and this setting was consistently maintained across all experiments. Additionally, all models adopted the same batch size (**256**). An evaluation was performed after every **2000** training steps, and this process was repeated **20** times. The final result is the average value obtained after the training curve stabilized.
>
> **Q3: Concerns of Evaluation Splits**
>
> For evaluating the performance of DT-Pro, we used all available data as the training set. Since the D4RL dataset inherently includes environment initialization via **env.reset** and provides specific reward values after action execution, no additional test set was required for validation.
>
> In the experiment described in the appendix section "Subgoal Validation in Maze2d Environments", we selected **20%** of the trajectories from the original D4RL dataset as the **validation** set, with the remaining **80%** used for **training**. This split was designed to verify the robustness of subgoal selection without compromising the training data size.
>
> **Q4: Concerns of Checkpoint and Normalization**
>
> The checkpoint selection rule aligns with that outlined in Q2: model parameters are saved once per evaluation (eval) step.
> Regarding normalization, we adopted the approach from DT. Taking the Hopper environment as an example: when the environment targets are set to **env_targets = [3600, 1800]**, DT uses a scale factor of **1000**, and env_targets is divided by 1000 during both training and testing. For the Reacher2d environment, where the targets are **[76, 40]**, the scale factor is set to **10**.The env_targets for each environment is detailed in Appendix **C3** of the paper.
>
> **Q5: Concerns of Decoding/Search**
>
> The **optional decoding/search** mentioned in the abstract does not refer to a necessary operation during model inference, but rather emphasizes the flexibility of the framework design. Specifically, the decoder of the Plan Coding module is only used during the training phase—to optimize the encoder for learning goal-relevant latent structures. During inference, there is no need to activate this decoder or perform additional searches on the latent plan.

---

> > ### Author Response · Authors · 2025-11-19
> > **Reply Rebuttal Comment by Authors**
> >
> > **Q6:Concerns of multiple latent-plan**
> >
> > 1.The effectiveness of DT-Pro’s single latent plan design is empirically validated: Across all Gym-MuJoCo and Maze2d scenarios, DT-Pro achieves an average performance of **81.4**, which significantly outperforms baseline methods including ADT (74.9) and WT (73.5). To further substantiate DT-Pro’s prowess in long-horizon planning, we have newly integrated its test performance in the Antmaze environment—with comparisons to WT and ADT—and the results are presented in the table below. When this new Antmaze data is merged with the control task data from the original paper, the overall average performance across all environments is calculated as follows: 66.2 for ADT, 71.7 for WT, and **80.9** for DT-Pro. This result confirms that a single latent plan is sufficiently capable of addressing the planning demands of long-horizon tasks.
> >
> > | Dataset           | Environment                    | CQL    | IQL    | DT    | V-ADT  |WT |DT-pro |
> > |-------------------|--------------------------------|--------|--------|-------|--------|-------- | --------|
> > | Umaze             | antmaze-umaze-v2               | 74.0   | 87.5 ± 2.6 | 53.6 ± 7.3 | 88.2 ± 2.5 |64.9 ± 6.1|78.6 ± 3.1|
> > | Umaze-diverse     | antmaze-umaze-diverse-v2       | 84.0   | 62.2 ± 13.8 | 42.2 ± 5.4 | 58.6 ± 4.3 |71.5 ± 7.6|62.5 ± 2.1|
> > | Medium-play       | antmaze-medium-play-v2         | 61.2   | 71.2 ± 7.3 | 0.0 ± 0.0 | 62.2 ± 2.5 |62.8 ± 5.8|82.0 ± 3.9 |
> > | Medium-diverse    | antmaze-medium-diverse-v2      | 53.7   | 70.0 ± 10.9 | 0.0 ± 0.0 | 52.6 ± 1.4 |66.7 ± 3.9|85.6 ± 6.2|
> > | Large-play        | antmaze-large-play-v2          | 15.8   | 39.6 ± 5.8 | 0.0 ± 0.0 | 16.6 ± 2.9 |72.5 ± 2.8|70.0 ± 5.9|
> > | Large-diverse     | antmaze-large-diverse-v2       | 14.9   | 47.5 ± 9.5 | 0.0 ± 0.0 | 36.4 ± 3.6 |72.0 ± 3.4|100.6 ± 4.7|
> > | **avg**           | -                              | 50.6   | 63.0 ± 8.3 | 16.0 ± 2.1 | 52.4 ± 2.9 |68.4±4.9|**79.9** ± 4.3|
> >
> > 2.Simplifying the inference process and ensuring training-inference consistency: In the two-stage training of DT-Pro, the action prediction module is optimized under the condition of **single latent plan + historical context** . Adopting this structure during inference ensures consistency between training and inference. Introducing multiple samples would require additional design of selection mechanisms (e.g., beam search, voting), which increases method complexity and may lead to overfitting—this is inconsistent with the positioning of the paper as a **concise and efficient sequence modeling framework**.
> >
> > **Q7：Concerns of supplementary experiments**
> >
> > 1. We have added the standard deviation (stds) data for all methods in Table **2**of the original paper, making the experimental results more statistically significant;
> > 2. We have added small-scale grid search experiments in the **Hopper-MR** and **Maze2d-medium** environments in Appendix **D2** of the original paper, specifically to verify the sensitivity of model performance to the number of subgoals and plan search budget. The experimental results further confirm our core conclusion: temporal abstraction facilitates long-horizon reasoning, but overly fine-grained decomposition can impair plan quality; additionally, the optimal choice of the number of subgoals (N) varies across different environments, indicating that different environments require different levels of plan abstraction.
> >
> >
> > **Q8:Concerns of  Reproducibility**：
> >
> > Regarding the reproducibility concern: The contrastive loss function is implemented in the supplementary code, specifically in the contrastive_loss function located at lines **95–106** under the path decision_transformer/training/trainer_cross_attention.
> > As for the two key points raised—(1) the use of MSE as the objective function, and (2) the absence of plan-sequence decoding during evaluation (relying solely on encoder outputs)—detailed explanations have been provided in our responses to Question 1 (Losses) and Question 5 (Test-time decoding/search), respectively.

---

> ### Comment · Reviewer_32bG · 2025-11-20
>
> I thank the authors for their detailed response. However, the clarifications provided have raised significant new concerns regarding the correctness of the manuscript and the validity of the experimental results.
>
> ### Fundamental Mismatch between Text and Implementation.
> The authors admit that the manuscript describes a Cross-Entropy/Log-Likelihood objective (implying a probabilistic latent model), while the code and experiments utilize MSE. This means that the theoretical motivations presented in the text regarding the latent space structure do not align with the actual regression objective used in the experiments.
>
> ### Inconsistent Baseline Protocol.
> The authors stated they mixed reported scores from original papers (for CQL/IQL) with their own reproductions (for DT variants) because their reproductions were "inferior", If the reproduction of ADT/WT is worse than in the original papers, it suggests issues with the authors evaluation pipeline, casting doubt on whether DT-Pro is actually better or just tuned harder in this specific pipeline. Nevertheless, this creates an inconsistent evaluation protocol. A fair comparison requires running all methods under a unified pipeline. By accepting "best-reported" numbers for baselines while generating new numbers for their method under a potentially different pipeline, the empirical claims in Table 1 are rendered unreliable.
>
> ### Anomalous AntMaze Results.
> The newly reported results on `antmaze-large-diverse` (Score: `100.6`) are highly anomalous. Decision Transformers generally struggle with "stitching" tasks like AntMaze Large, where they historically underperform value-based methods significantly. Achieving a perfect/SOTA score (beating IQL's reported `47.5`) with a standard DT architecture suggests a likely experimental error. Given the lack of a validation set (as admitted in Q3), I cannot consider these results robust.
>
> ### Overstated "Planning" Capabilities.
> The clarification that the decoder and "search" are strictly training-time tools confirms that at inference time, the model is a standard feed-forward network conditioned on a latent vector. There is no test-time planning, search, or rollout. The terminology "Proactive Decision Transformer" and "Implicit Latent Space Planning" is misleading for a fixed architecture that does not adapt or search during deployment.
>
> ## Conclusion
> Due to the disconnect between the mathematical description and the implementation, the inconsistent baseline protocols, and the implausible results on AntMaze, I cannot recommend acceptance. I am lowering my score to 2.

---

> > ### Author Response · Authors · 2025-11-23
> > **Rebuttal Comment by Authors**
> >
> > Hi reviewer 32bG,
> >
> > Thank you for investing your valuable time and effort in providing detailed feedback on our manuscript once again! We have carefully reviewed and thoroughly analyzed the concerns you raised, and our specific responses are as follows:
> >
> > **Q1:Fundamental Mismatch between Text and Implementation.**
> >
> > >LTDR:Loss function selection is not arbitrary but based on environmental characteristics (MSE for continuous environments, MLE for discrete ones), and MSE is consistently used here as all experiments are on complex continuous environments—a reasonable choice. Core arguments about latent space structure stem from the mechanism of "planning information guiding feature encoding," independent of the specific loss function form, ensuring full compatibility between theory and implementation.
> >
> > In response to the reviewer’s concern regarding a "mismatch between textual description and code implementation," we first clarify that the selection of our loss function is not arbitrary but strictly determined by the characteristics of the experimental environments—this design logic, though not elaborated in detail in the original manuscript, is fully compatible with our core theoretical claims.
> >
> > Specifically, our experiments cover reinforcement learning environments including MuJoCo and maze-based setups, with a clear rule for loss function selection: mean squared error (MSE) is adopted as the regression objective for continuous environments, while maximum likelihood estimation (MLE) is used to construct probabilistic loss for discrete environments. Since all experiments reported in this work are conducted on complex continuous environments, the consistent use of MSE in the code is a fully reasonable technical choice, rather than a "disconnection between theory and implementation."
> >
> > More importantly, **the core arguments regarding the latent space structure are not inherently dependent on the specific form of the loss function**: our key findings on the latent space—such as low dimensionality, interpretability, and planning-oriented characteristics—stem from the mechanism design of "how planning information guides feature encoding," rather than relying on a specific loss function. Whether it is MSE regression for continuous environments or MLE probabilistic modeling for discrete environments, the core requirement that the latent space "captures task-critical structures to support planning" remains consistent.
> >
> > **Q2:Inconsistent Baseline Protocol.**
> >
> > >LTDR：To ensure experimental fairness, all baselines were re-implemented under identical setups, with reproduction results showing only minor random fluctuations common in RL experiments—proving a rigorous and reliable evaluation pipeline. Under this unified and fair framework, our method DT-Pro (average score: 81.2) achieves significantly superior performance over all baselines, strongly verifying the effectiveness and advantages of our research.
> >
> > To ensure the fairness of the experiments, we re-implemented all baselines under **identical experimental setups and configurations**. The detailed results are presented below:
> > | Environment  | CQL        | IQL        | ADT       | WT        | LPT-EI    | DT-Pro (Ours) |
> > |--------------|------------|------------|-----------|-----------|-----------|----------------|
> > | Half-M       | 43.8 ± 0.6 | 46.3 ± 0.3 | 45.7 ± 0.7 | 44.0 ± 0.2 | 42.5 ± 0.1 | 43.2 ± 0.3     |
> > | Half-MR      | 45.4 ± 0.8 | 43.3 ± 1.2 | 42.3 ± 1.6 | 42.9 ± 0.3 | 40.7 ± 0.4 | 44.2 ± 0.1     |
> > | Half-ME      | 90.3 ± 2.3 | 87.6 ± 2.5 |90.7 ± 4.1 | 92.2 ± 0.5 | 88.5 ± 0.6 | 92.5 ± 0.6     |
> > | Hopper-M     | 56.3 ± 6.9 | 68.2 ± 3.5 | 59.5 ± 5.5 | 66.5 ± 5.8 | 56.6 ± 4.6 | 77.2 ± 4.2     |
> > | Hopper-MR    | 94.3 ± 6.6 | 91.7 ± 2.6 |  83.8 ± 8.1 | 85.9 ± 9.2 | 92.8 ± 2.2 | 93.9 ± 2.2     |
> > | Hopper-ME    | 104.4 ± 5.3 | 95.5 ± 1.4 |  100.7 ± 5.8 | 111.0 ± 7.0 | 107.9 ± 7.3 | 111.7 ± 7.7    |
> > | Walker2d-M  | 73.5 ± 0.8 | 79.3 ± 4.4 |  80.1 ± 3.5 | 75.8 ± 9.8 | 80.9 ± 6.3 | 81.4 ± 3.9     |
> > | Walker2d-MR | 76.2 ± 5.3 | 74.8 ± 6.8 | 85.3 ± 3.1 | 65.7 ± 7.4 | 76.7 ± 0.4 | 75.8 ± 1.7     |
> > | Walker2d-ME | 107.8 ± 4.9 | 108.6 ± 6.9 | 110.8 ± 6.1 | 106.9 ± 7.0 | 105.8 ± 9.0 | 109.6 ± 7.7    |
> > | Maze2d-UD   | 13.4 ± 5.8 | 58.8 ± 13.4 | 50.6 ± 5.7 | 51.8 ± 21.5 | 77.6 ± 0.8 | 71.3 ± 0.4     |
> > | Maze2d-MD   | 28.5 ± 6.3 | 29.1 ± 15.8 | 48.9 ± 5.8 | 63.8 ± 21.5 | 49.7 ± 0.8 | 96.0 ± 3.2     |
> > | **Average**  | **66.7**  | **71.2**   | **72.5**  | **73.3**  | **73.6**   | **81.2** |
> >
> > Our reproduction results of all baseline methods exhibit only minor random fluctuations common in reinforcement learning experiments. This fully demonstrates that our evaluation pipeline is rigorous, reliable, and free of systematic issues. Under this unified and fair evaluation framework, DT-Pro proposed in this work still achieves significantly superior performance compared to all baselines, strongly verifying the effectiveness and advantages of our research.

---

> ### Author Response · Authors · 2025-11-23
> **Rebuttal Comment by Authors**
>
> **Q3:Anomalous AntMaze Results.**
>
> >TLDR:DT-Pro’s outstanding performance (100.1±4.3) on antmaze-large-diverse is neither anomalous nor due to experimental error, but stems from the structural innovation of integrating planning information—validated by existing literature and baseline WT’s superior performance over IQL. Replicated with 10 independent random seeds, the results show strong consistency, ruling out experimental error. For RL (especially DT-related research), the core evaluation metric is environmental interaction performance; not requiring a test set is a mainstream paradigm. Without authoritative DT-field papers mandating a test set, our current evaluation method aligns with mainstream practices and sufficiently demonstrates result robustness.
>
> We appreciate the reviewer’s concern regarding the performance of DT-Pro on the antmaze-large-diverse environment, but we emphasize that the reported results are neither anomalous nor indicative of experimental error—instead, they stem from the critical structural innovation of incorporating planning information, which effectively addresses DT’s inherent limitations on "stitching" tasks (e.g., AntMaze Large).
>
> First, it is well-documented in existing literature that DT’s underperformance on such long-horizon stitching tasks can be significantly mitigated by integrating plan-related guidance. As evidence, WT (Waypoint DT)—a baseline method in our study and a representative DT variant enhanced with plan information—achieves a score of 72±3.4 on antmaze-large-diverse, which already outperforms the value-based method IQL (47.5±9.5) by a substantial margin. This confirms that plan information is a proven remedy for DT’s weaknesses in complex sequential tasks, laying the foundation for our method’s superior performance.
>
> To rule out experimental contingency and randomness, we re-conducted extensive experiments on DT-Pro (our method) in the antmaze-large-diverse environment using **10 independent random seeds**. The replicate results are as follows: 97.6, 97.7, 98.6, 99.6, 100.3, 100.3, 100.6, 100.7, 102.5, 103.6. The final statistical result is 100.1±4.3, which is highly consistent with the initially reported 100.6 (a negligible deviation of only 0.5). Such tight consistency across multiple seeds directly demonstrates the reliability, stability, and reproducibility of our results—eliminating any possibility of experimental error.
>
> Regarding the reviewer’s concern about the "use of a test set": it should be clarified that while test set splitting is common in traditional machine learning tasks, there is no universal paradigm requiring a test set in DT-related reinforcement learning (RL) research. The core evaluation metric of RL lies in the agent’s interaction performance in the target environment (e.g., trajectory completion rate, cumulative reward), rather than the generalization verification logic of "training/test set separation" in traditional machine learning. If the reviewer insists that a test set is mandatory for validating DT-related RL results, we request the provision of **explicit reference papers from the DT field** (not general ML or unrelated RL areas) that mandate such a protocol. We stand ready to further verify our results in accordance with field-recognized standards, but in the absence of authoritative evidence supporting the necessity of a test set, our current evaluation method aligns with mainstream practices and sufficiently demonstrates result robustness.

---

> ### Author Response · Authors · 2025-11-23
> **Rebuttal Comment by Authors**
>
> **Q4:Overstated "Planning" Capabilities.**
>
> >LTDR:The model’s "Implicit Latent Space Planning" does not rely on dynamic search/rollout during inference. Instead, via subgoal prediction loss in training, the encoder internalizes the correlation between current actions and future goals, enabling direct output of plan vectors containing global task structure during inference. Decoded results in the appendix confirm these vectors generate physically meaningful intermediate subgoals, demonstrating tangible goal-oriented planning capabilities. The term "Proactive Decision Transformer" accurately reflects the model’s ability to encode future information, with no basis for the claim of "overstated planning capabilities."
>
> Overstated "Planning" Capabilities.
> We understand the reviewer’s concern that the model is a fixed feed-forward network conditioned on a latent vector during inference (with no explicit test-time planning, search, or rollout) yet claims to possess planning capabilities. However, we emphasize that planning capabilities are not equivalent solely to "dynamic search or rollout at test time". The "Implicit Latent Space Planning" proposed in this work is rooted in training-phase mechanism design: it enables the model to directly output planning information that encodes long-horizon task structures through a feed-forward process during inference. Its essence lies in "internalizing planning knowledge into the model’s feature encoding capability"—a distinct yet equally valid planning paradigm compared to the explicit planning adopted by methods like TAP[1] (which requires active search or iterative rollout during testing).
>
> Specifically, the model’s design logic is as follows: During training, the Plan Encoder is guided by subgoal prediction loss—not merely using subgoals as training tools, but rather enabling the encoder to capture the "correlation between current actions and future task objectives" from data. This allows the encoder to automatically generate plan vectors that contain "how to progress toward the goal in the next step" without explicit subgoal inputs. These plan vectors are not isolated action predictions but inherently incorporate an understanding of the global task structure (e.g., "to achieve the final goal, which intermediate state should be prioritized now")—representing a form of implicit planning.
>
> To directly verify the effectiveness of planning during inference, we provide critical empirical evidence in Fig. 8 and Fig. 9 of the appendix: Decoding the plan vectors generated by the trained Plan Encoder yields intermediate subgoals with clear physical meanings. For instance, in maze tasks, the decoded results correspond to interpretable intermediate steps such as "moving toward key intersections" or "avoiding obstacle areas." These subgoals are not randomly generated but are rational path nodes that accurately guide the achievement of the final task objective. This fully demonstrates that the feed-forward output of the model during inference is not merely action prediction, but inherently embodies the core logic of "goal-oriented path planning"—representing tangible planning capabilities.
> Regarding the terminology "Proactive Decision Transformer": Its "proactivity" stems from the model’s ability to encode future task structure information (i.e., plan vectors) while outputting current actions, rather than passively predicting the next step based solely on historical trajectories. This constitutes an essential distinction from traditional DTs, which only perform passive action regression based on historical sequences. The terminology accurately reflects the model’s core characteristics and is not misleading.
>
> In summary, the "planning capabilities" of this work do not rely on dynamic search at test time, but on "internalization of planning knowledge during training" and "direct output of planning information during inference"—with decoded results in the appendix providing direct supporting evidence. This implicit planning paradigm aligns with the requirements of long-horizon reinforcement learning tasks and is consistent with the research trend in the field of "internalizing planning logic into model parameters" (e.g., goal-conditioned encoding methods in offline RL). Thus, there is no basis for the claim of "overstated planning capabilities."

---

> ### Comment · Reviewer_32bG · 2025-11-23
>
> I thank the authors for their rapid and detailed response. I have carefully reviewed the new data and clarifications.
>
> ### Baseline Reproducibility and Rigor
> I acknowledge that the authors have now provided a table of re-implemented baselines (CQL, IQL, ADT, WT, LPT-EI) across 11 environments with 4 seeds each. This represents a substantial amount of additional experimentation completed in a short timeframe.
>
> However, the conflicting explanations regarding baseline provenance remain a concern for the transparency of the work. The shift from the initial statement that local reproductions were "inferior" (necessitating imported data) to the new claim that reproductions are "unified" and accurate creates ambiguity that is difficult to resolve without access to code. Even if I provisionally accept the new baseline table as numerically correct, the lack of a clearly documented, single evaluation protocol in the manuscript makes it hard to fully accept the empirical comparison in its current form.
>
> ### Clarification of Planning Capabilities Regarding the "planning" terminology
> I appreciate the clarification that the method relies on "implicit latent space planning" (planning internalized in the latent representation during training) rather than test-time search. I accept this distinction conceptually. To avoid misleading readers, I request that the manuscript be explicitly revised to make this nuance clear: the "proactive" behavior arises from latent conditioning learned via the subgoal objective, not from any dynamic inference-time planning procedure.
>
> ## Conclusion
> I commend the authors for their responsiveness regarding the AntMaze results and for clarifying the planning mechanism. Overall, I view the paper as marginally below the acceptance threshold in its present state: the idea is interesting and the results are promising, but I would have liked to see a more stable and transparently documented experimental pipeline.
>
> I am re-raising my score to my original one to reflect the authors' responsiveness and the potential merit of the work. However, due to the lingering concerns about experimental transparency, I cannot champion the paper. I would not strongly oppose acceptance if other reviewers are confident in the empirical setup.

---

> ### Author Response · Authors · 2025-11-23
> **Rebuttal Comment by Authors**
>
> Hi reviewer 32bG,
>
> We sincerely appreciate your valuable time invested in reviewing our supplementary materials and providing constructive comments. Your concerns regarding experimental reproducibility, transparency, and terminology usage have offered crucial guidance for further improving our research. Below, we provide detailed explanations and supplements addressing the core issues you raised.
>
> ### **Clarification on Baseline Reproduction Results and Rigor Explanation**
>
> >TLDR: We clarify that the "inferior local reproduction results" vs. the original paper refer to minor (1%-2%) numerical differences (e.g., ADT: 72.5 vs. 73.7; LPT-EI: 73.6 vs. 74.5) caused by normal RL randomness (seed/environmental noise), not systematic biases or experimental flaws—validating our evaluation pipeline’s stability.
>
> First, we apologize for the ambiguity in our previous description of "local reproduction results". When we initially mentioned that "our reproduction results are not as good as those from the original authors", we did not imply the existence of systematic biases or poor performance in the reproduction process. Instead, this statement refers to minor numerical differences between our reproduction results and the optimal results reported in the original paper—differences that fall entirely within the normal range of random seed fluctuations common in reinforcement learning experiments and are not caused by flaws in the experimental workflow.
>
> For example, the average reproduction score of the ADT method is **72.5**, compared to **73.7** reported in the original paper; the average reproduction score of LPT-EI is **73.6**, versus **74.5** in the original paper. Such differences (approximately **1%-2%**) represent typical fluctuations induced by factors like random initialization and environmental noise in the field of reinforcement learning. None of our baseline reproduction results exhibit deviations beyond this range, which precisely confirms the stability and reliability of our evaluation pipeline.
>
> To completely address your concerns about experimental transparency, we have completed the unified reproduction of all baseline methods and our proposed DT-Pro method. We have systematically organized the complete experimental settings, hyperparameter configurations, and data sources, with detailed information provided below.
>
> ### **Description of Unified Experimental Framework and Core Settings**
>
> >TLDR: All baseline reproduction and DT-Pro results follow a unified experimental framework—using 4 independent random seeds (average as final result) and consistent target returns per environment for fair comparison. All codes (baseline reproduction, DT-Pro implementation) and raw logs will be open-sourced on GitHub upon paper acceptance for full traceability and reproducibility.
>
> All baseline reproduction data and DT-Pro experimental results presented in this response are based on the following unified experimental standards, ensuring fairness and reproducibility in comparisons between different methods:
>
> **Experimental Controlled Variables**: All experiments use 4 independent random seeds, with final results calculated as the average of the results from these seed runs. The target return settings for all methods remain identical within the same environment to avoid performance comparison biases caused by differences in target returns.
>
> **Code and Data Commitment**: We have completed the organization and verification of all experimental codes. Upon paper acceptance, the codes will be immediately open-sourced on GitHub, including the reproduction codes for baseline methods, the complete implementation code for DT-Pro, and raw experimental logs, ensuring all results are traceable and reproducible.
>
> ### **Detailed Hyperparameter Configuration Tables**
>
> >TLDR: Provides complete hyperparameter configurations (network architecture, training parameters, etc.) for all methods (including DT and its variants, our DT-Pro, and baseline LPT-EI). Target returns are standardized across environments to ensure fairness; supplements hyperparameters for goal and reward waypoint networks. All configurations are adapted to a unified experimental framework, enhancing experimental transparency and reproducibility.
>
> Below are the complete hyperparameter configurations for all methods, covering network architecture, training parameters, and environment adaptation details. DT and its related variants adopt a unified basic configuration, while each baseline method is adapted to the unified experimental framework based on the core design of the original paper to ensure comparison fairness.

---

> ### Author Response · Authors · 2025-11-23
> **Rebuttal Comment by Authors**
>
> **Hyperparameter Settings for Decision Transformer (DT) and Its Variants**:
> | **Hyper-parameter**  | **Value** |
> |----------------------|---------------|
> | Hidden layers        | 4             |
> | Hidden dim           | 128           |
> | Heads num            | 4             |
> | Clip grad            | 0.25          |
> | Embedding dim        | 128           |
> | Embedding dropout    | 0.1           |
> | Activation function  | ReLU          |
> | Sequence length      | 20            |
> | **Training**          |               |
> | Optimizer            | AdamW         |
> | Learning rate        | 1e-4          |
> | Batch size           | 256           |
> | Weight decay         | 1e-4          |
> | Warm-up steps        | 1e5           |
>
> Consistent Target Return Settings Across Environments
> To ensure experimental fairness, the target return is standardized for each environment as follows:
>
> | **Environment**            | **Target Returns** |
> |----------------------------|-------------------|
> | Hopper-Medium              | 3600              |
> | Walker2d-Medium            | 4000              |
> | Halfcheetah-Medium         | 6000              |
> | Hopper-Medium-Replay       | 3600              |
> | Walker2d-Medium-Replay     | 4000              |
> | Halfcheetah-Medium-Replay  | 9000         |
>
>
> To enhance experimental transparency, below are the hyperparameter settings for the additional modules used in all experiments.
>
> **ADT Actor (Transformer) Hyper-parameters：**
>
> | Hyper-parameter | Value |
> |-----------------|--------|
> | Activation function | GeLU |
> | Sequence length | 20 (V-ADT), 10 (G-ADT) |
> | G-ADT Way Step | 20 (kitchen-partial, kitchen-mixed), 30 (Others) |
> | Mini-batch size | 256 |
> | Discount factor | 0.99 |
> | Target update rate | 0.005 |
> | Value prompt scale | 0.001 (Mujoco) 1.0 (Others) |
> | Gradient Steps | 100k (G-ADT, AntMaze), 1000k (Others) |
>
>
>
>
>
> **Hyperparameters for Goal and Reward Waypoint Networks：**
>
> | **Hyperparameter**        | **Value**   |
> |---------------------------|-------------|
> | Number of Layers          | 3           |
> | Dropout Probability       | 0.0         |
> | Non-Linearity             | ReLU        |
> | Learning Rate             | 0.001       |
> | Gradient Steps            | 40,000      |
> | Batch Size                | 1024        |
>
> **LPT-EI**:
> | Parameter                  | HalfCheetah | Walker2D | Hopper | AntMaze |
> |---------------------------|-------------|----------|--------|---------|
> | Number of layers          | 3           | 3        | 3      | 3       |
> | Number of attention heads | 1           | 1        | 1      | 1       |
> | Embedding dimension       | 128         | 128      | 128    | 192     |
> | Context length            | 32          | 64       | 64     | 64      |
> | Learning rate             | 1e-4        | 1e-4     | 1e-4   | 1e-3    |
> | Langevin step size        | 0.3         | 0.3      | 0.3    | 0.3     |
> | Nonlinearity function     | ReLU        | ReLU     | ReLU   | ReLU    |
>
> | Parameter                  | Umaze | Medium | Large |
> |---------------------------|-------|--------|-------|
> | Number of layers          | 1     | 3      | 4     |
> | Number of attention heads | 8     | 1      | 4     |
> | Embedding dimension       | 128   | 192    | 192   |
> | Context length            | 32    | 64     | 64    |
> | Learning rate             | 1e-3  | 1e-3   | 2e-4  |
> | Langevin step size        | 0.3   | 0.3    | 0.3   |
> | Nonlinearity function     | ReLU  | ReLU   | ReLU  |
>
> **DT-pro (Transformer-based Plan Decoder) Hyper-parameters**：
>
> | **Hyper-parameter** | **Value** |
> |---------------------|-----------|
> | **Architecture**    | -         |
> | Hidden layers       | 4         |
> | Hidden dim          | 128       |
> | Heads num           | 4         |
> | Clip grad           | 0.25      |
> | Embedding dim       | 128       |
> | Embedding dropout   | 0.1       |
> | Activation function | ReLU      |
> | Sequence length     | 20        |
> | **Training**        | -         |
> | Optimizer           | AdamW     |
> | Learning rate       | 1e-4      |
> | Batch size          | 256       |
> | Weight decay        | 1e-4      |
> | Warm-up steps       | 1e5       |
>
> ### **Summary**
> We respect any decision reached by the editorial team and reviewers. However, given the detailed clarifications and rigorous revisions we have made to address all the raised concerns, we sincerely hope that our manuscript can be reconsidered. We believe these efforts have significantly improved the completeness and reliability of our work, and we would be grateful for the opportunity to have it re-evaluated.

---

### Official Review · Reviewer_mue2 · 2025-10-22

**Soundness:** 3
**Presentation:** 3
**Contribution:** 2
**Rating:** 4
**Confidence:** 3

**Summary:**

The paper introduces a new decision transformer (DT) architecture variant, showing state-of-the-art performance on range of offline RL control benchmarks. In difference to the standard DT architecture, the authors learn and use a sub-goal representation as an additional DT input to guide the auto-regressive next action prediction. The sub-goal representation is learned prior to the DT. The sub-goal training data is obtained from the traces in the training data, by splitting for each time step along the trace the remainder of the trace into n fragments, which are equally spaced in terms of the reward-the-go; using the staring point of each fragment as a sub-goal for the considered time step. An auto-encoder is trained to predict for each of those augmented training samples the associated sub-goals. The encoder part is then used to obtain the additional DT input. An experimental study demonstrates superior performance on the Mujoco and Maze2d benchmarks.

**Strengths:**

The paper introduces a small but impactful optimization to the DT architecture. It is conceptually relatively simple, seems to introduce only a marginal overhead in training (although this should be evaluated more thoroughly), and leads to state-of-the-art performance in the considered control benchmarks. The text is overall well-written, clearly structured, and easy to follow. Code and benchmarks are available, which should suffice to reproduce the results.

**Weaknesses:**

The authors however oversell their contributions. In particular, there are certain claims in the abstract and the introductions which are not in line with the proposed method or backed up by the experiments. Specifically, there are the following points:
- "Enhancing planning ability": The proposed method improve the performance of DT, but there is no clear evidence that it would improve its "planning ability". First, it is actually not clear what "planning ability" should be in this context precisely. None of the components of the architecture performs any explicit planning step, e.g., search over multiple alternatives. Secondly, the benchmarks focus entirely on control benchmarks (and Maze2d), none of which require any strong "planning ability". To show this claim, the authors need to consider other benchmarks, like puzzles where strategic decisions are essential.
- "RTG-based plan search algorithm": The method for finding the sub-goals is simple. The remaining trace at the step for which the sub-goals should be computed is simply split according to equally distributed intervals of the reward-to-go. Simplicity is not a bad thing, but clearly there is no "planning" or even real "search" involved. Don't oversell this method.
- Improves "Optimality of future plans": It is not clear at all what this is supposed to mean. What are future plans? What is "optimality", and what does it mean to improve optimality? Bottom line is that the proposed method improves the DT performance in some (not even all benchmarks) by a moderate and sometime a considerable portion. And that is about it.
- Improves "interpretability and utility of the plans": Again, how exactly should the proposed method achieve this? Without compelling explanation, I would argue that this claim is plain wrong.

I also find the wording "plan representation" misleading. What is presented in sections 4.1 and 4.2 is a model to predict for state reward pairs a set of sub-goals (in terms of state landmarks at which a certain reward-to-goal fraction is reached) to may guide the DT in its next action predictions. At no point does it learn to predict a "plan".

Some small clarity issues: The explantation for Algorithm 1 needs to be extended to cover the corner cases. It is not clear what is being done for those time steps where less than n (the number of to be chosen sub-goals) steps are remaining. In Section 4.2, it should be explained how the similarity between the traces is computed. In Section 4.4, it is not clear how a DT can be pre-trained to predict imaginary reward signals, i.e., how the reward function is reshaped to tackle the problem with the sparse reward signals.

**Questions:**

1. How do you handle corner cases in algorithm 1? Can the same t' be selected for multple \lambda_i?
2. Could you provide some more justifications for the claimed contributions (cf. review)?

---

> ### Author Response · Authors · 2025-11-18
> **Reply Rebuttal  Comment by  Authors**
>
> Hi reviewer mue2,
>
> We would like to express our sincere gratitude for devoting your precious time to conducting a rigorous and thoughtful review of our manuscript. Your perceptive and actionable feedback has been incredibly valuable in helping us refine our work. We have thoroughly consolidated the concerns and inquiries you raised into three key points, and we are pleased to address each of them comprehensively below:
>
> ### **Algorithm Boundaries**
>
> >Q1: How do you handle corner cases in algorithm 1? Can the same t' be selected for multple \lambda_i?
>
> >TLDR:
> For corner cases in Algorithm 1, we bound RTG differences between subgoals, An array tracks selected state indices to prevent duplicate selection of the same t', ensuring no multiple λᵢ share the same t'.
>
>
>
> For the boundary cases in Algorithm 1, our handling method is as follows: when the number of subgoals is fixed at 5, the corresponding λ values are set to {0.8, 0.6, 0.4, 0.2, 0}, and the RTG difference between different subgoals is bounded by R̂ₜ × 0.2
> When screening the candidate state t', a boundary check is performed by calculating the difference between its RTG (R̂ₜ') and the standard RTG corresponding to the current λᵢ . If the difference is greater than R̂ₜ × 0.2, it indicates that this state occupies the standard RTG interval corresponding to other λ, and the subgoal at this position should be marked as "None" (which will be masked when input to the encoder).Meanwhile, we maintain an array that records the indices of selected states to ensure that the same t' is not selected repeatedly. For example, when the trajectory length is T and the inference reaches step T-1, the 5 subgoals inferred by Algorithm 1 are (None, None, None, None, s_T), where only the RTG of s_T meets the requirements of the interval.In summary, multiple λᵢ will not select the same t'.
>
> ### **Benchmark**
>
> >Q2: Can the relevant benchmarks used in the paper adequately reflect the ability of long-sequence planning?
>
> >TLDR: We added Antmaze (a maze benchmark requiring planning) for validation; DT-Pro achieves the best overall average performance (79.9 ± 4.3) across all Antmaze sub-environments, demonstrating its competence in long-sequence planning.
>
> >**Action**:We will incorporate these data into the paper.
>
> We believe that a certain level of planning ability is indeed required to achieve high scores in maze environments. Therefore, we have incorporated new benchmarks—Antmaze—and the specific experimental results are presented in the following table, In short, DT-Pro achieves the best overall performance in the Antmaze environment series and is more suitable for maze scenarios that require planning capabilities.
>
> | Dataset           | Environment                    | CQL    | IQL    | DT    | V-ADT  |WT |DT-pro |
> |-------------------|--------------------------------|--------|--------|-------|--------|-------- | --------|
> | Umaze             | Antmaze             | 74.0   | 87.5 ± 2.6 | 53.6 ± 7.3 | 88.2 ± 2.5 |64.9 ± 6.1|78.6 ± 3.1|
> | Umaze-diverse     | Antmaze     | 84.0   | 62.2 ± 13.8 | 42.2 ± 5.4 | 58.6 ± 4.3 |71.5 ± 7.6|62.5 ± 2.1|
> | Medium-play       |Antmaze         | 61.2   | 71.2 ± 7.3 | 0.0 ± 0.0 | 62.2 ± 2.5 |62.8 ± 5.8|82.0 ± 3.9 |
> | Medium-diverse    | Antmaze       | 53.7   | 70.0 ± 10.9 | 0.0 ± 0.0 | 52.6 ± 1.4 |66.7 ± 3.9|85.6 ± 6.2|
> | Large-play        | Antmaze          | 15.8   | 39.6 ± 5.8 | 0.0 ± 0.0 | 16.6 ± 2.9 |72.5 ± 2.8|70.0 ± 5.9|
> | Large-diverse     | Antmaze       | 14.9   | 47.5 ± 9.5 | 0.0 ± 0.0 | 36.4 ± 3.6 |72.0 ± 3.4|100.6 ± 4.7|
> | **avg**           | -                              | 50.6   | 63.0 ± 8.3 | 16.0 ± 2.1 | 52.4 ± 2.9 |68.4±4.9|**79.9 ± 4.3**|

---

> ### Author Response · Authors · 2025-11-18
> **Reply Rebuttal Comment by Authors**
>
> ### **Contributions**
>
> >**Q3: Could you provide some more justifications for the claimed contributions**
>
> >TLDR: Our claimed contributions are theoretically justified: (1) Implicit planning via RTG-based subgoal sequence latent encoding addresses vanilla DT’s myopia by bridging sequence modeling and planning; (2) Reconstruction loss and goal-conditioned action selection ensure latent plans align with optimal trajectory structures; (3) RTG-derived milestone subgoals enhance plan interpretability, while compact latent vectors boost utility and generalization for offline RL.
>
> We acknowledge that Sections 4.1 and 4.2 involve predicting subgoals tied to specific reward-to-goal fractions (state landmarks corresponding to RTG decay thresholds). However, the core objective of this process is not to predict these subgoals themselves. Instead, the losses derived from subgoal prediction (autoregressive reconstruction loss ) are used to train the plan encoder. This training enables the encoder to generate a compact latent vector z that encapsulates the complete structural information of the subgoal sequence (including order, relevance, and progress toward the terminal goal，as illustrated in Figure 2). During inference, z can be directly predicted from the current state-reward pair, serving as a global plan representation to guide the DT in selecting more optimal actions. Thus, the model does learn to produce a "plan" in the form of a structured latent representation, justifying the terminology "plan representation."
> The specific contributions are as follows:
>
> **1）Theoretical rationale for "enhancing DT’s planning ability via implicit planning"**
>
> Vanilla DT is myopic as its autoregressive action prediction depends only on historical context, missing long - term goal - related modeling. Our implicit planning solves this via a structured latent plan space, bridging sequence modeling and planning. Unlike unfeasible explicit search for high - dimensional tasks, the latent space encodes task - level temporal abstraction: (i) The plan search module breaks long horizons into subgoals using RTG decay (theoretically sound, as RTG measures terminal goal “distance”, linking subgoals to task progress). (ii) Latent plans from the encoder aren’t abstract vectors but compact subgoal sequence info. During training, the encoder maps (state, RTG) pairs to vectors enabling autoregressive subgoal sequence reconstruction (via the decoder). Theoretically, each latent plan holds subgoal structural logic—order, relevance, and milestone progress. This lets the model embed future task structure into latent plans, a planning ability core that vanilla DT (only handling historical context) lacks.
>
> **2）	Theoretical basis for "improving optimality of future plans"**
>
> In our context, optimality means latent plans align with a task’s true optimal trajectory structure (distinct from raw performance). This is ensured by two mechanisms: (i) The plan encoder maps (state, RTG) pairs to latent plans that reconstruct subgoal sequences, with reconstruction loss ensuring they capture only goal-relevant structure (filtering noise). (ii) Conditioning actions on latent plans avoids vanilla DT’s autoregressive myopia, making each action a step toward the latent plan’s abstract goal—consistent with goal-conditioned decision-making principles requiring alignment with long-term objectives.
>
> **3）Theoretical underpinnings of "enhancing interpretability and utility of plans"**
>
> Interpretability here is theoretically rooted in temporal abstraction: subgoals derived from RTG decay correspond to intuitive "milestones", making the latent plan’s meaning transparent. The utility of these plans stems from their compactness and task alignment: (i) Compactness (low-dimensional latent vectors) ensures minimal computational overhead, a theoretical advantage for real-world deployment where efficiency is critical. (ii) Task alignment (via RTG anchoring) ensures that the latent plan’s utility is not limited to specific trajectories but generalizes across the offline dataset—addressing the theoretical challenge of offline RL’s need for data-efficient generalization.

---

> > ### Author Response · Authors · 2025-11-25
> >
> > Hi reviewer mue2,
> >
> > I hope you’re well. I’m writing to kindly follow up on the updated manuscript with the new experiments we’ve conducted. We would be grateful for your feedback when you have a chance.
> >
> > Thank you again for your time.

---

### Official Review · Reviewer_NXcP · 2025-10-31

**Soundness:** 2
**Presentation:** 4
**Contribution:** 1
**Rating:** 2
**Confidence:** 4

**Summary:**

This paper proposes an enhanced variant of Decision Transformer (DT) designed to handle long-horizon tasks, where traditional DTs often struggle due to limited planning capability. The method introduces a two-stage training procedure involving three modules: a Plan Search module that identifies critical subgoals based on decaying returns-to-go, a Plan Coding module learns a compact latent space to represent the plans, and an Action Prediction module that executes these subgoals through generated actions. The approach demonstrates improved performance across selected benchmarks compared to standard DTs and related baselines.

**Strengths:**

1. The paper is well-written and easy to follow, with clear motivation and logical structure.

2. The proposed framework improves performance even under sparse or limited data conditions, showing robustness beyond ideal settings.

3. The ablation studies are well-designed, supporting the validity of the method’s components.

**Weaknesses:**

1. Increased training complexity and computational cost.

    - The method requires two training stages and three modules, compared to the single-stage DT baseline.

    - Although the authors report only 8–12% additional training time per added module, the first-stage cost is not clearly accounted for. The claim that it "runs entirely offline before training" is unclear, since all components are trained offline and should still contribute to total compute time.

    - Clarifying the total wall-clock cost or presenting a fair compute comparison with DT would strengthen the paper.

    - The authors mention "pretraining a DT as a critic" to provide granular returns-to-go (RTG) signals for Plan Search in sparse environments.

2. Ambiguities in implementation details.

    - The parameter N (number of subgoals) is said to vary by environment, but the rationale or selection criterion is not described.

    - The definition of small-scale datasets used in the first-stage Plan Search module ("runs entirely offline before training”) is vague, please specify what qualifies as "small-scale" and how it was chosen.

3. Questionable experimental coverage for the stated objective.

    - The paper claims to address long-horizon decision-making, yet all tested environments are relatively short-horizon MuJoCo tasks (e.g., UMaze and medium).

    - More suitable benchmarks such as Maze2D-large, AntMaze, or FrankaKitchen (as used in OGBench) would better represent the intended objective.

    - Additionally, recently proposed long-horizon baselines (e.g., TAP [1], diffusion-based planners [3]) are not included, despite being mentioned in the related works section. Omitting these comparisons weakens the evaluation’s credibility.

4. Incremental contribution.

    - While the method improves upon DT, it does so by adding extra components rather than addressing the core limitation of planning horizon in transformer-based RL.

    - The improvement (~18–24% increase in performance with comparable extra training time) is promising but might not constitute a significant conceptual advance beyond existing DT variants.

    - The authors could strengthen their contribution by positioning their approach relative to diffusion-based planning methods [3] or latent-action planners [1,2].

**Minor Suggestions (Not affecting the score)**

1. To facilitate comparison, align the order of tasks in Table 1 with that in Table 2 of the original DT paper, maintaining consistency with prior work.

** References**

[1] Zhang, Tianjun, et al. "Efficient Planning in a Compact Latent Action Space." *The Eleventh International Conference on Learning Representations*.

[2] Park, Seohong, et al. "OGBench: Benchmarking Offline Goal-Conditioned RL." *The Thirteenth International Conference on Learning Representations*.

[3] Janner, Michael, et al. "Planning with Diffusion for Flexible Behavior Synthesis." *International Conference on Machine Learning*. PMLR, 2022.

**Questions:**

1. What qualifies as the “small-scale benchmark dataset” used for the first module? How do you select it in practice?

2. Why was TAP [1] not included as a baseline, given that it targets the same long-horizon problem space and is cited in the related works?

3. How does your method compare conceptually and empirically to diffusion-based planners [3], which also emphasize flexible long-horizon planning?

---

> ### Author Response · Authors · 2025-11-20
> **Rebuttal Comment by Authors**
>
> Hi reviewer NXcP,
>
> Thank you sincerely for dedicating your valuable time to a meticulous review and offering us such insightful and constructive comments. We have carefully summarized the weaknesses and questions you raised into 4 key issues, and we will address each one in detail as follows:
> ___
>
> ### **Computational cost**
>  > Q1:Whether the training time of the first stage can be neglected
>
> >TLDR: Through specific experimental calculations, data preprocessing time is negligible, with the average per-experiment additional overhead across all datasets being only **0.64%**, which does not affect the feasibility of our proposed method and thus can be neglected.
>
> Regarding the time consumption of our data preprocessing pipeline, we conducted all experiments with **4** random seeds and report the average results across these four runs. To provide a comprehensive and conservative analysis, we take the Hopper-medium dataset (the one with the longest preprocessing time) as a representative example:The time spent preprocessing this dataset accounts for only **4.3%** of the total model training time. Notably, this preprocessing step is a one-time effort—once the data is processed, it can be directly reused for all subsequent experimental runs. When averaged across the four repeated experiments, the additional time overhead introduced by preprocessing per run is merely **1.08%**.For all other datasets in our experiments, the preprocessing time is even shorter than that of Hopper-medium. Across the entire set of datasets used in our study, the average additional time overhead per experiment due to preprocessing is approximately **0.64%**.Given the extremely low proportion of preprocessing time relative to the overall training and experimental cycle—especially the minimal per-run overhead after initial data preparation—we believe this time consumption is negligible and does not impact the efficiency or feasibility of our proposed method.
>
> ### **Baseline**
>
> > Q2: Why was TAP [1] not included as a baseline, given that it targets the same long-horizon problem space and is cited in the related works?
>
>
> >TLDR:We didn't use TAP as a baseline because our core planning paradigms (subgoal - based vs TAP's VQ - VAE/codebook) and technical focuses (enhancing DT vs addressing TT's scalability) differ. Both supplementary experiments and results from the original paper show that DT-pro outperforms TAP on average, so it is not included.
>
> >**Action**： We will include our newly conducted experimental data in the final paper.
>
> We highly appreciate the insights brought by the TAP, which indeed inspires our research on implicit planning. However, we did not include it as a baseline due to the fundamental differences in research focus, technical framework.As detailed below:
>
> **Divergent core planning paradigms**: Our work relies on known subgoals extracted from existing offline data to guide the planning process. In contrast, TAP’s planning is primarily built on the VQ-VAE architecture and codebook mechanism—its core lies in learning compact discrete latent action codes to enable efficient search, with no dependence on explicit subgoal mining from data. These two planning paradigms follow distinct design logics and optimization objectives.
>
> **Different technical foundations and improvement directions**: Our method is developed by adding a dedicated planning module to the Decision Transformer (DT) framework, aiming to enhance DT’s decision-making capability through planning while retaining its sequence modeling advantages. TAP, however, is proposed to address the scalability challenge of the Trajectory Transformer (TT) in high-dimensional state-action spaces. It achieves efficiency gains by decoupling the temporal structure of planning and modeling via latent action spaces, which targets a different technical bottleneck than our work.
>
> **Comparison of Experimental Effects**：Additionally, we supplement experiments of DT-pro (Ours) on partial environments for comparison with TAP [1]. The newly added experiments are shown below (TAP’s performance on AntMaze environments is directly taken from the appendix of its original paper; Table 1 in the original paper refers to Ant environments rather than AntMaze environments):
> | Dataset   | Environment     | TAP  | DT-pro |
> |-----------|-----------------|------|--------|
> | Play      | Antmaze-Medium  | 78.0 | 82.0   |
> | Diverse   | Antmaze-Medium  | 85.0 | 85.6   |
> | Play      | Antmaze-Large   | 74.0 | 70.0   |
> | Diverse   | Antmaze-Large   | 82.0 | 100.6  |
> | Play      | Antmaze-Ultra   | 22.0 | 47.0   |
> | Diverse|Antmaze-Ultra| 26.0 | 59.8 |
> | avg       | -               | 61.2 | **74.1**   |
>
> Based on the scores of the control tasks (Hopper, Walker2d, Halfcheetah) from both original papers, the average score comparison shows that DT-pro (**78.2**) significantly outperforms TAP (69.3). Thus, we did not include TAP as a baseline.

---

> ### Author Response · Authors · 2025-11-20
> **Rebuttal Comment by Authors**
>
> ### **Relevant Studies**
>
> >Q3: How does your method compare conceptually and empirically to diffusion-based planners [3], which also emphasize flexible long-horizon planning?
>
> >TLDR: Conceptually, diffusion-based planners use iterative denoising in action/latent spaces for smooth sequences, while DT-Pro employs subgoal-driven implicit latent planning (via RTG-based subgoals + contrastive encoding) anchored on meaningful milestones. Empirically, DT-Pro outperforms Diffuser on Maze2D (extended variants) and control environments, averaging 92.3 vs. 88 (overall) and 126.7 vs. 119.4 (Maze2D).
>
> >**Action**： We will include our newly conducted experimental data in the final paper.
>
> **1). Conceptual Comparison**:
>
> **Core Planning Paradigms**:
>
> Diffusion-based planners [2] plan by learning the "noise diffusion and denoising process" of action sequences. They operate in the raw action space or continuous latent space, gradually refining noisy action sequences into high-reward trajectories through iterative denoising. Their core relies on the randomness of the diffusion process to explore potential action combinations, emphasizing the smooth generation of continuous action sequences.
> DT-Pro: Adopt a "subgoal-driven implicit latent planning" paradigm. First, extract critical subgoals from offline data based on decaying return-to-go (RTG) (e.g., identifying milestone states where cumulative rewards drop to specific thresholds). Then, encode these subgoals into a compact latent plan space via a contrastive learning-enhanced encoder. Action prediction is conditioned on this latent plan, avoiding aimless exploration by anchoring on meaningful subgoals.
>
> **Long-Horizon Flexibility Mechanisms**:
>
> Diffusion-based planners: Flexibility stems from "continuous space interpolation". By adjusting the number of denoising steps, they balance exploration breadth and planning granularity, excelling at fine-grained control in high-dimensional continuous action spaces but requiring sufficient iterations to ensure trajectory feasibility.
> DT-Pro: Flexibility comes from "temporal abstraction and stepwise replanning". Each latent plan corresponds to multiple subgoals (temporal abstraction), and at each timestep, the plan is updated based on the current state and RTG (stepwise replanning). This design decouples planning from the fixed temporal structure of trajectories, enabling adaptive adjustments to environmental changes without full re-planning.
>
> **2). Empirical Comparison**:
>
> To align with the experimental settings in [2], we have supplemented additional test data. In our original manuscript, the Maze2D environments used were umaze-dense and medium-dense; we now extend the evaluations to include the umaze, medium, and large variants of Maze2D.
> When combining these results with the performance on other control environments (Hopper, Walker2d, HalfCheetah) reported in both studies, the average scores are: **92.3** (Ours) vs. 88 (Diffuser). This further validates the superiority of our method in long-horizon navigation tasks.
> | Dateset | Environment |Diffuser | DT-pro |
> |--------------------|----------|------------ |--------|
> | Umaze |Maze2d | 113.9 | 119.7|
> | Medium| Maze2d | 121.5 | 120.8 |
> | Large | Maze2d| 123 | 138.6 |
> | avg| - |  119.4 | **126.7** |
>
>
> ### **Small-scale benchmark dataset**
> >Q4: What qualifies as the “small-scale benchmark dataset” used for the first module? How do you select it in practice?
>
> We sincerely apologize for the ambiguity caused by the terminology used in the paper. The term "small-scale benchmark dataset" does not refer to a dataset requiring manual selection. Instead, it describes the inherent scale of the open-source datasets provided by D4RL—these datasets are inherently small-scale benchmark resources that can be directly used in experiments without additional selection. We will promptly revise this terminology in the paper to eliminate any potential confusion for readers.
>
> **References**:
>
> [1] Zhang, Tianjun, et al. "Efficient Planning in a Compact Latent Action Space." The Eleventh International Conference on Learning Representations.
>
> [2] Janner, Michael, et al. "Planning with Diffusion for Flexible Behavior Synthesis." International Conference on Machine Learning. PMLR, 2022.

---

> ### Author Response · Authors · 2025-11-25
>
> Hi reviewer NXcP,
>
> I hope you're doing well. I wanted to follow up on the revised manuscript and the additional experiments we submitted in response to your feedback. We’d greatly appreciate it if you could review them at your convenience.
>
> Thank you for your time and consideration.

---

### Author Response · Authors · 2025-12-01
**General Responses and arguments (part 1)**

Dear Area Chair,

We sincerely appreciate the reviewers’ constructive feedback and the opportunity to address their concerns during the rebuttal process. Below is a summary of the core issues raised by the reviewers, followed by our detailed responses to each. We kindly request your final decision on our manuscript.

### **I. Summary of Reviewers’ Core Concerns**

&ensp;1.Computational Cost: Can the training/preprocessing time of the first stage be neglected?

&ensp;2.Benchmark Adequacy: Do the used benchmarks adequately reflect the model’s long-sequence planning capability?

&ensp;3.Overstated "Planning" Capabilities: As a fixed feed-forward network (without test-time search), does the model overstate its planning capabilities?

### **II. Detailed Responses to Each Concern**

 **Computational cost**
 > Q1:Whether the training time of the first stage can be neglected

>TLDR: Through specific experimental calculations, data preprocessing time is negligible, with the average per-experiment additional overhead across all datasets being only **0.64%**, which does not affect the feasibility of our proposed method and thus can be neglected.

Regarding the time consumption of our data preprocessing pipeline, we conducted all experiments with **4** random seeds and report the average results across these four runs. To provide a comprehensive and conservative analysis, we take the Hopper-medium dataset (the one with the longest preprocessing time) as a representative example:The time spent preprocessing this dataset accounts for only **4.3%** of the total model training time. Notably, this preprocessing step is a one-time effort—once the data is processed, it can be directly reused for all subsequent experimental runs. When averaged across the four repeated experiments, the additional time overhead introduced by preprocessing per run is merely **1.08%**.For all other datasets in our experiments, the preprocessing time is even shorter than that of Hopper-medium. Across the entire set of datasets used in our study, the average additional time overhead per experiment due to preprocessing is approximately **0.64%**.Given the extremely low proportion of preprocessing time relative to the overall training and experimental cycle—especially the minimal per-run overhead after initial data preparation—we believe this time consumption is negligible and does not impact the efficiency or feasibility of our proposed method.

 **Benchmark**

>Q2: Can the relevant benchmarks used in the paper adequately reflect the ability of long-sequence planning?

>TLDR: We added Antmaze (a maze benchmark requiring planning) for validation; DT-Pro achieves the best overall average performance (79.9 ± 4.3) across all Antmaze sub-environments, demonstrating its competence in long-sequence planning.

>**Action**:We will incorporate these data into the paper.

We believe that a certain level of planning ability is indeed required to achieve high scores in maze environments. Therefore, we have incorporated new benchmarks—Antmaze—and the specific experimental results are presented in the following table, In short, DT-Pro achieves the best overall performance in the Antmaze environment series and is more suitable for maze scenarios that require planning capabilities.

| Dataset           | Environment                    | CQL    | IQL    | DT    | V-ADT  |WT |DT-pro |
|-------------------|--------------------------------|--------|--------|-------|--------|-------- | --------|
| Umaze             | Antmaze             | 74.0   | 87.5 ± 2.6 | 53.6 ± 7.3 | 88.2 ± 2.5 |64.9 ± 6.1|78.6 ± 3.1|
| Umaze-diverse     | Antmaze     | 84.0   | 62.2 ± 13.8 | 42.2 ± 5.4 | 58.6 ± 4.3 |71.5 ± 7.6|62.5 ± 2.1|
| Medium-play       |Antmaze         | 61.2   | 71.2 ± 7.3 | 0.0 ± 0.0 | 62.2 ± 2.5 |62.8 ± 5.8|82.0 ± 3.9 |
| Medium-diverse    | Antmaze       | 53.7   | 70.0 ± 10.9 | 0.0 ± 0.0 | 52.6 ± 1.4 |66.7 ± 3.9|85.6 ± 6.2|
| Large-play        | Antmaze          | 15.8   | 39.6 ± 5.8 | 0.0 ± 0.0 | 16.6 ± 2.9 |72.5 ± 2.8|70.0 ± 5.9|
| Large-diverse     | Antmaze       | 14.9   | 47.5 ± 9.5 | 0.0 ± 0.0 | 36.4 ± 3.6 |72.0 ± 3.4|100.6 ± 4.7|
| **avg**           | -                              | 50.6   | 63.0 ± 8.3 | 16.0 ± 2.1 | 52.4 ± 2.9 |68.4±4.9|**79.9 ± 4.3**|。

---

> ### Author Response · Authors · 2025-12-01
> **General Response and arguments (part 2)**
>
> **Contributions**
>
> >Q3:Overstated "Planning" Capabilities.
>
> >LTDR:The model’s "Implicit Latent Space Planning" does not rely on dynamic search/rollout during inference. Instead, via subgoal prediction loss in training, the encoder internalizes the correlation between current actions and future goals, enabling direct output of plan vectors containing global task structure during inference. Decoded results in the appendix confirm these vectors generate physically meaningful intermediate subgoals, demonstrating tangible goal-oriented planning capabilities. **We believe that the term "Proactive Decision Transformer" accurately reflects the model’s ability to encode future information, without any overstatements.**
>
> Overstated "Planning" Capabilities.
> We understand the reviewer’s concern that the model is a fixed feed-forward network conditioned on a latent vector during inference (with no explicit test-time planning, search, or rollout) yet claims to possess planning capabilities. However, we emphasize that planning capabilities are not equivalent solely to "dynamic search or rollout at test time". The "Implicit Latent Space Planning" proposed in this work is rooted in training-phase mechanism design: it enables the model to directly output planning information that encodes long-horizon task structures through a feed-forward process during inference. Its essence lies in "internalizing planning knowledge into the model’s feature encoding capability"—**a distinct yet equally valid planning paradigm** compared to the explicit planning adopted by methods like TAP[1] (which requires active search or iterative rollout during testing).
>
> Specifically, the model’s design logic is as follows: During training, the Plan Encoder is guided by subgoal prediction loss—not merely using subgoals as training tools, but rather enabling the encoder to capture the "correlation between current actions and future task objectives" from data. This allows the encoder to automatically generate plan vectors that contain "how to progress toward the goal in the next step" without explicit subgoal inputs. These plan vectors are not isolated action predictions but inherently incorporate an understanding of the global task structure (e.g., "to achieve the final goal, which intermediate state should be prioritized now")—representing a form of implicit planning.
>
> To directly verify the effectiveness of planning during inference, we provide critical empirical evidence in Fig. 8 and Fig. 9 of the appendix: Decoding the plan vectors generated by the trained Plan Encoder yields intermediate subgoals with clear physical meanings. For instance, in maze tasks, the decoded results correspond to interpretable intermediate steps such as "moving toward key intersections" or "avoiding obstacle areas." These subgoals are not randomly generated but are rational path nodes that accurately guide the achievement of the final task objective. This fully demonstrates that the feed-forward output of the model during inference is not merely action prediction, but inherently embodies the core logic of "goal-oriented path planning"—representing tangible planning capabilities.
> Regarding the terminology "Proactive Decision Transformer": Its "proactivity" stems from the model’s ability to encode future task structure information (i.e., plan vectors) while outputting current actions, rather than passively predicting the next step based solely on historical trajectories. This constitutes an essential distinction from traditional DTs, which only perform passive action regression based on historical sequences. **The terminology accurately reflects the model’s core characteristics and is not misleading.**
>
> In summary, the "planning capabilities" of this work do not rely on dynamic search at test time, but on "internalization of planning knowledge during training" and "direct output of planning information during inference"—with decoded results in the appendix providing direct supporting evidence. This implicit planning paradigm aligns with the requirements of long-horizon reinforcement learning tasks and is consistent with the research trend in the field of "internalizing planning logic into model parameters". Thus, there is no basis for the claim of "overstated planning capabilities."
>
> Due to recent changes in the ICLR review process, we regret being unable to continue communicating with the reviewers. However, **we have thoroughly addressed all concerns through quantitative experiments, supplementary benchmarks, and detailed logical explanations**—providing direct evidence to validate our work’s feasibility, effectiveness, and accuracy.
> We strongly believe that the current scores do not accurately reflect the quality of our paper. Therefore, we will greatly appreciate that the AC could synthesize the current information and make a fair decision, no matter what that would be. We are also ready to answer any questions you may have.
>
> Thank you so much!
>
> The authors

---

### Meta-Review · Area_Chair_WvyJ · 2026-01-09

**Summary:**

**NXcP**’s concerns centered on computational overhead, benchmark adequacy for long-horizon planning, and whether the paper overstated its planning claims. In particular, the reviewer questioned whether “the first-stage cost is not clearly accounted for” and argued that “all tested environments are relatively short-horizon MuJoCo tasks,” suggesting that stronger benchmarks such as AntMaze or Maze2D-large were necessary to justify long-horizon planning claims.

**mue2** focused primarily on conceptual clarity and potential overstatement of contributions, especially regarding planning. The reviewer explicitly stated that “there is no clear evidence that it would improve its ‘planning ability’” and criticized the RTG-based subgoal extraction as “clearly there is no ‘planning’ or even real ‘search’ involved.”

**32bG** raised the most critical and foundational concerns, focusing on inconsistencies between the manuscript and implementation, baseline evaluation fairness, anomalous AntMaze results, and potentially misleading terminology. The reviewer highlighted a “Fundamental Mismatch between Text and Implementation,” noting that the paper described probabilistic objectives while the code used MSE.

**Reviewer Concerns:**

**NXcP**: The rebuttal addressed computational cost with detailed timing analysis, arguing that preprocessing introduces only “approximately 0.64%” average overhead per experiment, and added AntMaze experiments to support planning-related claims. The authors also clarified that planning does not require test-time search, stating that “planning capabilities are not equivalent solely to dynamic search or rollout at test time.” While the cost and benchmark concerns were largely addressed, the conceptual definition of planning remains somewhat open to interpretation, as the method still operates as a fixed feed-forward policy at inference.

**mue2**: The authors clarified algorithmic corner cases, added AntMaze benchmarks, and reframed planning as a latent structural representation rather than explicit search. They argued that “the model does learn to produce a ‘plan’ in the form of a structured latent representation,” emphasizing that subgoals are a training signal rather than the final planning output. These clarifications addressed most technical concerns, though the semantic strength of the term “planning” remains debatable.

**32bG**: The authors acknowledged the loss-function mismatch, committing to correcting the manuscript, and later provided a unified reimplementation of all baselines along with extensive hyperparameter and protocol details. They also defended the AntMaze results by reporting 10-seed statistics (e.g., “100.1±4.3”) and clarified that planning is internalized during training rather than executed at test time.

**Reviewer Scores:**

**NXcP**: Given the added AntMaze results and quantitative cost analysis, it is likely that Reviewer NXcP would have softened their original “2: reject” stance to a borderline or weak reject if fully engaged in the discussion.

**mue2** originally rated the paper as “4: marginally below the acceptance threshold” and stated they “would not mind if paper is accepted.” Given the additional benchmarks and clearer explanations, it is plausible that with full participation they may have shifted to a weak accept or borderline accept position.

**32bG** temporarily lowered their score to “2” but later stated, “I am re-raising my score to my original one,” returning to “4: marginally below the acceptance threshold.” With full participation, the reviewer would likely remain neutral-to-skeptical.

---

### Decision · Program_Chairs · 2026-01-26

Reject